Resource

# Protein and RNA ADP-ribosylation detection is influenced by sample preparation and reagents used

Lisa Weixler[1], Nonso Josephat Ikenga[1], Jim Voorneveld[2], Gülcan Aydin[1], Timo MHR Bolte[1], Jeffrey Momoh[1], Mareike Bütepage[1], Alexandra Golzmann[1], Bernhard Lüscher[1], Dmitri V Filippov[2], Roko Žaja[1], Karla LH Feijs[1]

The modification of substrates with ADP-ribose (ADPr) is important in, for example, antiviral immunity and cancer. Recently, several reagents were developed to detect ADP-ribosylation; however, it is unknown whether they recognise ADPr, specific amino acid–ADPr linkages, or ADPr with the surrounding protein backbone. We first optimised methods to prepare extracts containing ADPr–proteins and observe that depending on the amino acid modified, the modification is heatlabile. We tested the reactivity of available reagents with diverse ADP-ribosylated protein and RNA substrates and observed that not all reagents are equally suited for all substrates. Next, we determined cross-reactivity with adenylylated RNA, AMPylated proteins, and metabolites, including NADH, which are detected by some reagents. Lastly, we analysed ADP-ribosylation using confocal microscopy, where depending on the fixation method, either mitochondrion, nucleus, or nucleolus is stained. This study allows future work dissecting the function of ADP-ribosylation in cells, both on protein and on RNA substrates, as we optimised sample preparation methods and have defined the reagents suitable for specific methods and substrates.

## Introduction

The posttranslational modification of proteins is a well-known way to regulate proteins in response to changes in nutrient availability, viral infection, DNA damage, and many other signals. Proteins with catalytic activity can be switched on or off; others can change their localisation within the cell or interact with different molecules. ADP-ribosylation is a posttranslational modification, which is mediated in cells by ADP-ribosyltransferases (ARTs) of the ARTD family, which add ADP-ribose (ADPr) to their targets while releasing nicotinamide from co-factor NAD$^+$ (Gibson & Kraus, 2012; Luscher et al, 2018). Best studied is the modification with chains of ADPr,

termed poly(ADP-ribosyl)ation (PARylation), which amongst other processes has a demonstrated role in the DNA damage response, regulation of protein stability, and Wnt signalling. Inhibitors of the enzymes capable of generating PAR chains, the poly-ARTs, have found clinical applications in specific breast and ovarian cancers, with more clinical trials underway (Slade, 2020).

Much more enigmatic is its little sibling, the modification of proteins with just one ADPr moiety (MARylation). The ARTD family contains 17 members (Schreiber et al, 2006; Hottiger et al, 2010), of which only four are capable of generating chains of ADPr, namely, PARP1, PARP2, TNKS1, and TNKS2. The other members of this protein family have mono(ADP-ribosyl)transferase activity, or no proven activity yet in the case of PARP13 (Kleine et al, 2008; Vyas et al, 2014). Despite the potentially misleading name PARP for MARylating enzymes, this term was kept for historical reasons (Luscher et al, 2021). MARylation may play a role in immunity and transcription amongst other functions (Feijs et al, 2013c; Luscher et al, 2018; Fehr et al, 2020; Challa et al, 2021; Hopp & Hottiger, 2021b), which are very varied: PARP10 may regulate kinase activity (Feijs et al, 2013b; Rosenthal et al, 2013; Zhao et al, 2018) and replication (Schleicher et al, 2018); PARP12 localises to the Golgi and stress granules and may function there (Welsby et al, 2014; Catara et al, 2017); and PARP14 was reported not only to act as transcriptional co-activator (Goenka et al, 2007) but also to be present at focal adhesions (Vyas et al, 2013) and to be involved in DNA replication and repair (Nicolae et al, 2015; Dhoonmoon et al, 2020). Recent work on PARP7 points to a role in both the immunity and the regulation of the cytoskeleton and suggests that it may serve as an anti-cancer drug target (Gozgit et al, 2021; Palavalli Parsons et al, 2021; Rodriguez et al, 2021). These highly diverse functions of MARylation and associated transferases have been reviewed in detail (Feijs et al, 2013c; Luscher et al, 2018; Challa et al, 2021; Hopp & Hottiger, 2021b).

Different amino acids were identified as ADPr acceptors in recent years using different approaches, including mass spectrometry: glutamates, serines, tyrosines, histidines, and most recently cysteines (Buch-Larsen et al, 2020; Nowak et al, 2020; Gehrig et al, 2021;

---

[1]Institute of Biochemistry and Molecular Biology, RWTH Aachen University, Aachen, Germany  [2]Leiden Institute of Chemistry, Leiden University Department of Bioorganic Synthesis, Leiden, Netherlands

Correspondence: rzaja@ukaachen.de; kfeijs@ukaachen.de
Jeffrey Momoh's present address is Department of Nanomedicine and Theranostics, Institute for Experimental Molecular Imaging, Faculty of Medicine, RWTH Aachen University, Aachen, Germany

Rodriguez et al, 2021; Suskiewicz et al, 2021). In addition to the modification of proteins with ADPr, recent in vitro data indicate that some of the mammalian PARPs can also modify nucleic acids as summarised elsewhere (Groslambert et al, 2021; Weixler et al, 2021), although it is not clear yet what role RNA and DNA MARylation has in cells.

MARylation of both proteins and nucleic acids is a reversible modification, with different hydrolases removing the modification from different acceptor sites. ARH1 removes ADPr from arginine (Moss et al, 1992), and ARH3 from serine (Abplanalp et al, 2017), and MACROD1/MACROD2/TARG1 reverse the modification of acidic residues (Feijs et al, 2013a; Jankevicius et al, 2013; Rosenthal et al, 2013; Rack et al, 2020). Both PARG and ARH3 cleave the *O*-glycosidic bond between ribose moieties in PAR and are thus responsible for reversing PARylation (Rack et al, 2020). PARG leaves the terminal ADPr on the substrates, whereas ARH3 can remove it if linked to serine. No enzyme has been identified yet that reverses the ADP-ribosylation of cysteine residues. Not only mammalian genomes encode for macrodomain-containing proteins, but they are also present in other organisms (Rack et al, 2020). Triggered by the COVID-19 pandemic, the spotlight has recently been on the SARS-CoV-2 macrodomain protein Mac1, which was shown to be an ADP-ribosylhydrolase with thus far unknown substrates (Alhammad & Fehr, 2020; Fehr et al, 2020; Alhammad et al, 2021). Not only certain transferases but also a number of the mammalian hydrolases have been suggested to drive certain aspects of tumorigenesis, such as transformation, growth, and invasiveness (Feijs et al, 2020; Ishiwata-Endo et al, 2020). This has been difficult to verify, as antibodies for the hydrolases were poorly characterised and antibodies specific for MARylation were not available at all.

Research in the area of MARylation has been held back by the lack of tools for the analysis and the detection of modified substrates. In recent years, two major breakthroughs have occurred: optimised mass spectrometry methods, which are now reliably able to detect the modified proteins present in cells (Martello et al, 2016; Hendriks et al, 2019; Nowak et al, 2020); and the development of multiple antibodies and other reagents that detect this modification. This provides a great opportunity that has to be enjoyed with caution: most of the reported detection tools were tested on specific substrates, but their actual epitopes have been poorly mapped. It is thus not clear whether it is possible to directly compare one study performed with one antibody with another study using another detection tool. The first identified specialised readers of MARylation are the macrodomains of PARP14, which have been used to detect intracellular MARylation using live-cell imaging in the Lüscher laboratory (Forst et al, 2013; Butepage et al, 2018a). This was turned into a commercially available detection reagent by fusion of specific macrodomains to the Fc region of rabbit immunoglobulin by the Kraus laboratory, which is available from Millipore (Gibson et al, 2017). Next, an antibody was generated by Cell Signaling Technology against a peptide with a MARylated lysine, which appears to efficiently detect cellular MARylation and PARylation (Lu et al, 2019). This was followed by the creation of a modified version of the MAR-hydrolase Af1521 fused to an Fc-tag for detection by the Hottiger laboratory, which has increased binding affinity compared with the wild-type protein but is still catalytically active (Nowak et al, 2020). Next, the Matic laboratory generated

antibodies against MARylated serines and a general ADPr antibody, which are available from Bio-Rad (Bonfiglio et al, 2020). Lastly, the Hottiger laboratory raised a polyclonal antibody in rabbits, which detects both PAR- and MARylation (Hopp et al, 2021). Several reagents are thus available that can be used to study the in vivo function of MARylation; however, none of these reagents have been compared with each other. It is not clear whether there are differences in substrate recognition, and whether some may preferentially bind to specific MARylated amino acids or even recognise part of the protein backbone. It is highly probable that the MAR-specific reagents do recognise either the specific ADPr–protein bond or the part of the protein surrounding, as otherwise they would be expected to be efficient tools to detect PARylation as well.

We have compared the above-mentioned different reagents to map their respective specificities. For this purpose, we enzymatically generated protein substrates MARylated on specific amino acids, such as serine, arginine, and glutamate, or on peptides chemically modified on serine, cysteine, and threonine, and MARylated nucleic acid substrates, and tested all the detection reagents for ADP-ribosylation that were available to us. We verified the specific modification of our substrates using a panel of hydrolases, which reverse the modification only from specific targets as expected. We have included a detection reagent based on murine PARP14 macrodomains, which was previously described to have a higher affinity for ADPr than for the human macrodomains (Forst et al, 2013) and appears to be an efficient tool for immunoprecipitation of ADP-ribosylated nucleic acids. After determining the specificities of the antibodies on in vitro generated substrates, we next asked whether differences exist in their recognition of the modification introduced by different PARPs overexpressed in cells and tested their suitability for immunofluorescence using different fixation methods. To be able to do this, we optimised the preparation of cell lysates to ensure the maximum retention of the ADPr signal. Collectively, our work deciphers which reagents are suitable for which purpose and has optimised sample preparation procedures that allow the detection of low ADP-ribosylation levels.

## Results

### ADPr–detection reagents have different specificities in vitro

To be able to compare the currently available ADP-ribosylation detection reagents, we required suitable, well-defined substrates. Therefore, we assembled a collection of ARTs with known substrate specificity. Using these enzymes, we can generate proteins specifically modified on cysteine by pertussis toxin (PT), acidic amino acids by PARP10, arginine by mART2.2 and SpvB, serine by PARP1/HPF1, or PAR by PARP1. Recent studies reported the modification of other amino acids, such as histidine or tyrosine; however, the respective enzymes are not yet known and could therefore not be included. Most of the proteins employed here were purified from bacterial expression systems, with the exception of PARP1, which was immunoprecipitated using a GFP-tag from HEK293T cells (Fig 1A). All purified proteins are active and either automodify or ADP-ribosylate a specific target (Fig S1), with the exception of the

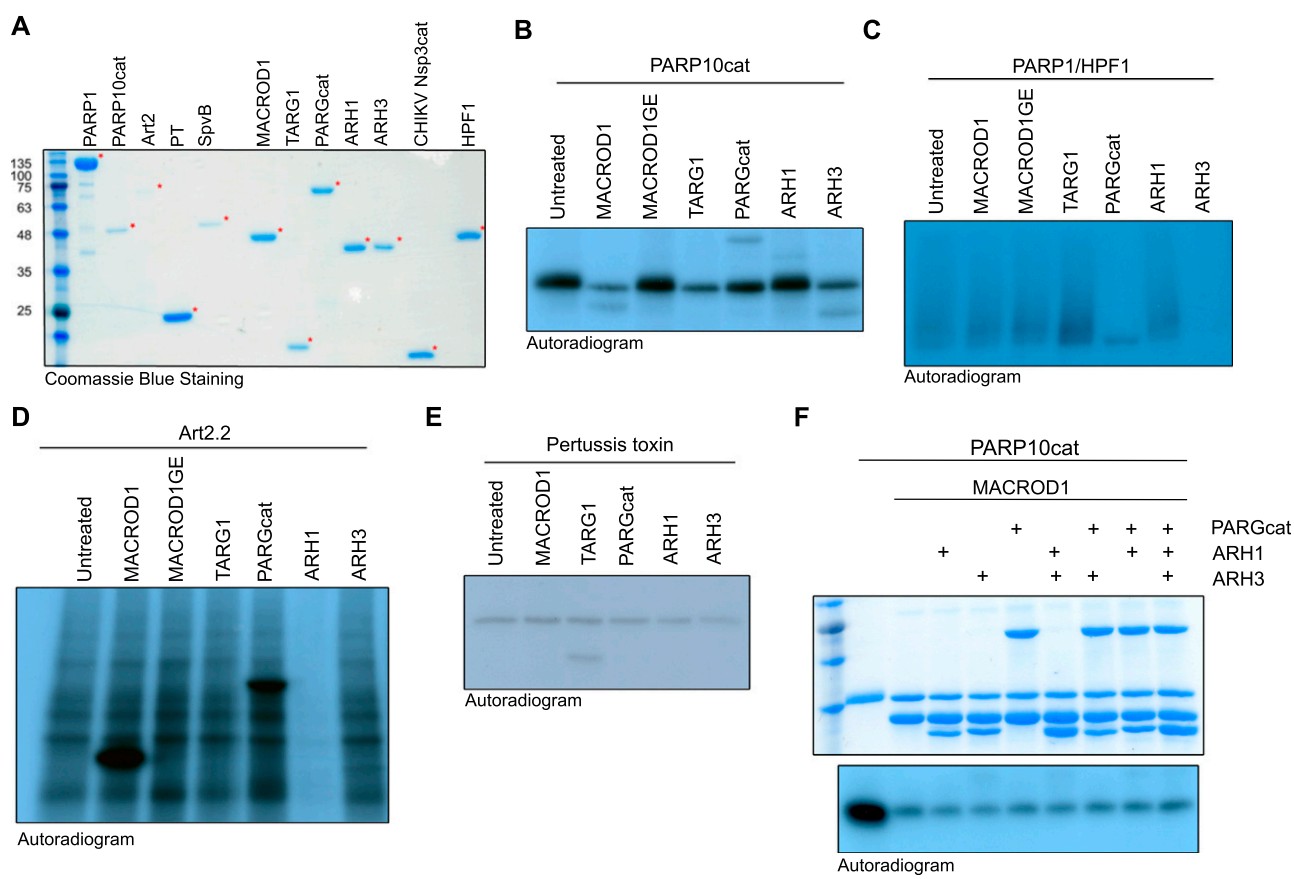

**Figure 1.  Enzymatic generation of specific ADP-ribosylated substrates.**
**(A)** Coomassie blue staining of the purified transferases, hydrolases, and co-factors used in this study. **(B, C, D, E)** Indicated transferases were incubated with $^{32}$P-NAD$^+$ and incubated at 37°C for 30 min. A cytoplasmic extract was provided to supply the mAART2.2 and pertussis toxin substrates. After the transferase reaction, OUL35 was added to inhibit PARP10 or olaparib was added to inhibit PARP1/HPF1, followed by a 30-min hydrolase reaction. Samples were run on SDS–PAGE, and incorporated radioactivity was visualised. The indicated MACROD1 mutant is an inactivating glycine 270 to glutamate mutation. **(F)** PARP10 was automodified using $^{32}$P-NAD$^+$, the reaction was stopped with OUL35, and indicated hydrolases were added. The Coomassie staining is displayed to show the relative amounts of hydrolases added. The samples were analysed as in (B).
Source data are available for this figure.

arginine-ART mART2.2, which modifies many proteins present in a cytosolic extract. Nuclei and mitochondria were removed as they contain the largest amounts of PARP1, TARG1, and MACROD1 and could confound the ADP-ribosylation assay. We noticed automodification also for PT, which is in line with previous observations using this truncated version of the protein (Ashok et al, 2020). To confirm the specificity of ADP-ribosylation reactions, we incubated these substrates with a panel of hydrolases. For PARP10 and PARP1/HPF1, we made use of specific inhibitors to stop the reactions before adding hydrolases (Venkannagari et al, 2016). ARH1 is only capable of reversing arginine modification, whereas ARH3 both hydrolyses PAR and removes ADPr linked to serine. PARG is only capable of hydrolysing the glycosidic bond between ADPr moieties and should not be able to remove the ADPr linked to the proteins. MACROD1, MACROD2, and TARG1 have been described to reverse the modification of glutamate (Jankevicius et al, 2013; Rosenthal et al, 2013). For the cysteine-linked ADP-ribosylation introduced by PT, no erasers are known to date. We purified the known erasers of ADP-ribosylation from bacteria: MACROD1, MACROD2, TARG1, PARG, ARH1,

and ARH3 (Fig 1A). Most of the described activities could confirm, for example, that MACROD1 and TARG1 reverse the modification introduced by PARP10 although not fully as has been seen before (Figs 1B and S2) (Kleine et al, 2008; Garcia-Saura & Schuler, 2021). We could furthermore confirm that ARH3 not only reverses both PAR and serine-linked MARylation (Fig 1C) but also appears to have some activity towards PARP10 (Fig 1B). This was hinted at in earlier work studying the reversal of ADP-ribosylation by PARP10 (Kleine et al, 2008) and would be expected if PARP10 also automodifies on serine as reported (Garcia-Saura & Schuler, 2021). ARH1 reverses the arginine modification introduced by mART2.2 very efficiently, but has no activity towards other modified amino acids (Fig 1D). As the hydrolases reverse the diverse modifications as expected, we concluded that our in vitro substrates are modified on the expected amino acids. None of the hydrolases were able to reverse the modification from cysteine as generated by PT (Fig 1E). This raises the possibility that additional mammalian intracellular hydrolases remain to be discovered. Alternatively, the automodification of the truncated PT we used may not represent an accessible substrate for

**Table 1. Reagents used in this study.**

| # | Reagent | Source | Type | Ref |
|---|---------|--------|------|-----|
| I | Anti-MAR | Millipore | PARP14-macro2/3 | Gibson et al (2017) |
| II | eAf1521-Fc | Michael Hottiger | Af1521 | Nowak et al (2020) |
| III | mPARP14-macro2/3-Fc | Bernhard Lüscher | mPARP14-macro2/3 | — |
| IIIm | mPARP14-macro2/3-GE-Fc | Bernhard Lüscher | mPARP14-macro2/3-GE | — |
| IV | Anti-MAR | Michael Hottiger | Polyclonal antibody | Hopp et al (2021) |
| V | Anti-PAR/MAR | Cell Signaling Technology | Monoclonal antibody | Lu et al (2019) |
| VI | Pan-ADPr | Ivan Matic | Monoclonal antibody | Bonfiglio et al (2020) |
| VII | Anti-PAR | Trevigen | Polyclonal antibody | — |

the hydrolases tested and we can thus not exclude that they are capable of reversing cysteine MARylation on other substrates.

In addition, we also noted that none of the hydrolases is able to completely reverse PARP10 modification and speculated that PARP10 is promiscuous and able to modify more than one type of amino acid. The partial reversal of PARP10 automodification by PARG and ARH3 was observed before, which may be the reversal of oligomers that PARP10 was reported to generate (Kleine et al, 2008). A more recent study using recombinant proteins also hinted at PARP10 promiscuity, as the modification of serine, arginine, and glutamate by PARP10 was detected using mass spectrometry (Garcia-Saura & Schuler, 2021). We incubated the protein with different hydrolases and combinations thereof. We were not able to further decrease the modification after the removal of most of the signal by MACROD1, as no other hydrolase leads to further decrease in the remaining signal (Fig 1F). Three potential explanations come to mind: first, PARP10 in addition modifies an amino acid for which we have not identified a hydrolase yet, such as cysteine; second, the signal is an artefact of the in vitro reaction and, for example, on the terminal amine; or third, the bond to the side chain of an acidic amino acid migrates from the C1′ to the C2′ or C3′ position and thus is no longer available to known hydrolases.

Having thus confirmed the specific modification of our enzymatically generated substrates, we next performed Western blots to test the specificities of the antibodies and detection reagents available to us. We generated a large amount of substrates (Fig S1), stored them at −20°C, and proceeded with Western blots once radioactivity decayed with the detection reagents listed in Table 1. We first used the macrodomain-based detection reagents on these substrates, Reagents I–III, which are based on either human or murine PARP14 macrodomains or contain the aforementioned Af1521 fused to an Fc (Fig 2A–D). We could confirm their reported specificity for MARylation over PARylation, as also, for example, the PARP1-HPF1 sample resulted in a more specific band instead of a smear. We developed the murine PARP14 macrodomain-based detection reagent, Reagent III or mPARP14-m2m3-Fc, as an affinity for ADPr was reported to be higher for the mouse than for the human macrodomain proteins (Forst et al, 2013; Butepage et al, 2018a). We also generated a control, Reagent IIIm or mPARP14-m2m3-GE-Fc, which has impaired ADPr binding. This mutant does not show these specific signals. Reagent I may detect some proteins independent of ADP-ribosylation status, as the higher molecular

weight species detected in the toxin lanes should not be present (Fig 2A). These toxins have very specific activities and do not modify proteins at these sizes. The polyclonal antibody Reagents IV (Fig 2E) and V (Fig 2F) have very similar properties on these in vitro substrates, although only Reagent IV detects in addition higher molecular weight species in the PARP10 sample. These could potentially be contaminants present in the reaction, which are modified by PARP10, as there is no substrate added to the reaction that should give a signal at this site. The pan-ADPr antibody Reagent VI shows the highest affinity for Arg-ADPr and Ser-ADPr (Fig 2G). We also tested an antibody that was described to detect PARylation, Reagent VII. It recognises PARylation as stated, but also for MARylation, bands are detectable (Fig 2H). This raises the question whether a fraction of the modification generated by these enzymes is oligomers, or whether the antibody can detect single ADPr moieties. It, for example, detects the modification introduced by PARP10, which could reflect a minor oligo-ADPr-transferase activity which PARP10 was reported to possess (Kleine et al, 2008). A significant background is present, despite identical processing of the blots, making this antibody potentially less suitable. Interestingly, the polymers are hardly detected by any of the antibodies in this experiment, although some in theory should. It is possible that the long storage in the freezer led to the degradation of these samples. We therefore performed new analyses of GFP-PARP1 immunoprecipitated from HEK293T cells. In parallel, we performed an experiment using radioactively labelled PAR chains, demonstrating the effective reversal of the oligomers by PARG with the exception of the last ADPr, whose resistance to reversal by MACROD1 confirms the specificity of this signal: oligo-ADPr attached to the substrate via serine (Fig S3). Reagent VI shows the weakest staining overall; the other tested reagents all detect PARylation and Ser-ADPr efficiently with the exception of Reagent III, which does not detect PARylation at all and detects Ser-ADPr relatively weakly (Fig 2I). To be able to test some substrates we cannot generate enzymatically, we next slot-blotted chemically synthesised peptides modified with ADPr on serine, cysteine, and threonine (Liu et al, 2019), including a peptide modified with phospho-ribose as a negative control, and tested the affinity of the different reagents towards these substrates (Fig 2J). The results are consistent with previous observations: Reagents III and VII lead to high background staining; the anti-ADPr antibody Reagents IV and V detect cysteine, threonine, and serine ADP-ribosylation, as does the reagent based

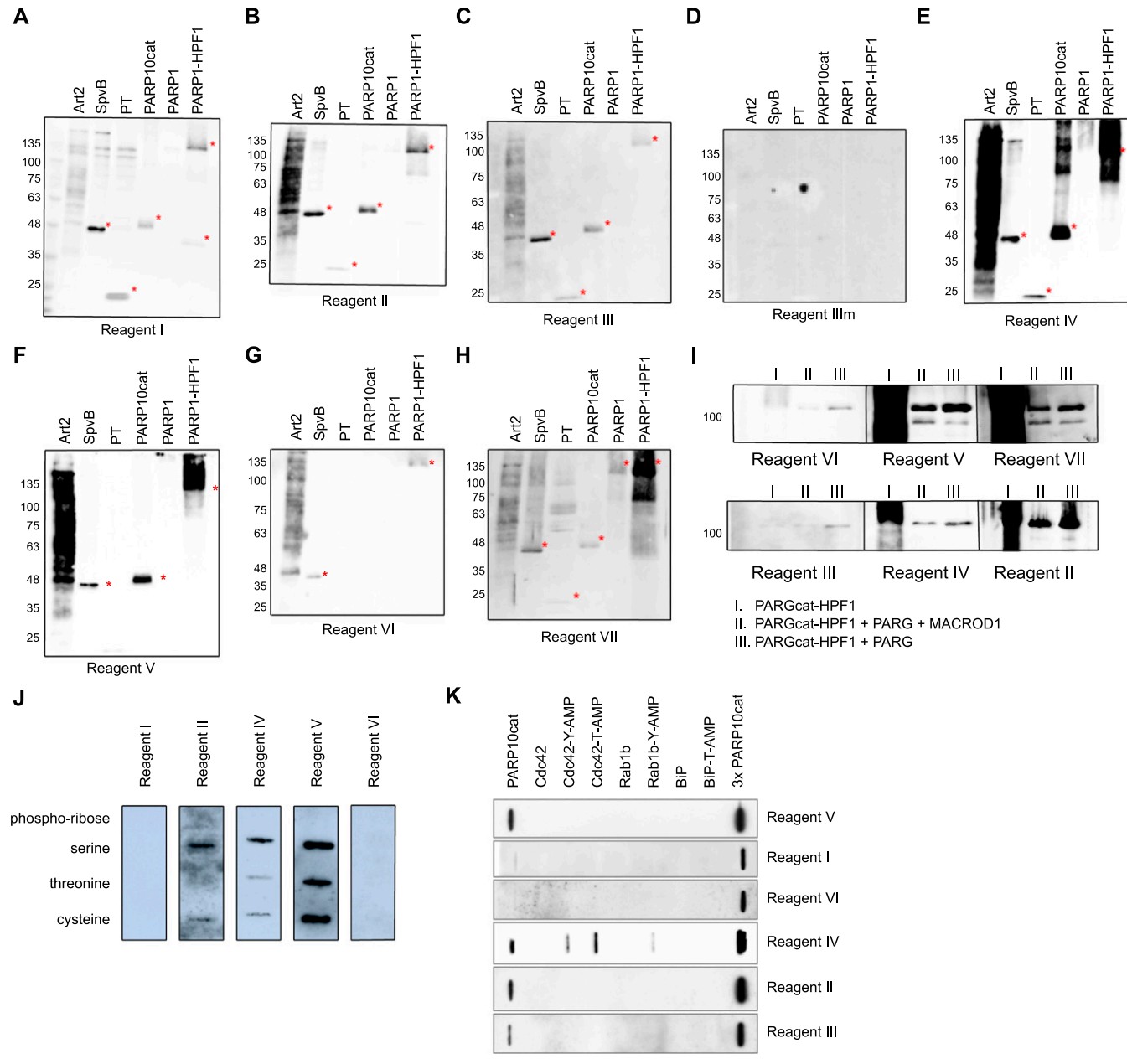

**Figure 2. Specificity of the reagents towards in vitro modified substrates.**
**(A, B, C, D, E, F, G, H)** ADP-ribosylation reactions from Fig S1 were loaded on multiple gels and blotted. Membranes were blocked with 5% milk in TBST and incubated with primary antibodies overnight. Asterisks indicate the transferases. **(I)** GFP-PARP1/His-HPF1 was incubated with NAD⁺ (I), followed by PARGcat and MACROD1 (II) or PARGcat (III) treatment. After blotting, the modification was detected using the same reagents. **(J)** 2 µM chemically synthesised peptide modified with either phospho-ribose or ADP-ribose on serine, threonine, or cysteine was slot-blotted and analysed using the indicated reagents. **(K)** 150 ng of AMPylated proteins was slot-blotted and analysed using the indicated reagents.
Source data are available for this figure.

on Af1521, Reagent II. The fact that both Reagent I and Reagent III, which are based on human and mouse PARP14 macrodomains, respectively, do not recognise any of these peptides implies that they do recognise the part of the protein surrounding the modification, which may not be present in the tested peptides. This offers a tentative explanation why these modules are specific for MARylated proteins: if they would detect ADPr or a part thereof, they

should at least detect the terminal ADPr of PAR chains as well. The binding of the module to the surrounding backbone would explain their exclusive detection of MAR- but not PARylated substrates. As none of the antibodies detect phospho-ribose, this does not appear to be part of the epitope, making it in theory possible that these reagents also detect proteins modified with AMP, which exists as posttranslational modification (Casey & Orth, 2018): those reagents

that are independent of amino acid linkage and surrounding amino acids may also detect proteins modified with AMP, as was demonstrated for a pan-ADPr reagent (Hopfner et al, 2020). We obtained and tested several AMPylated proteins using slot blot and observe that one of the ADPr reagents cross-reacts with AMPylation (Fig 2K), which could potentially lead to AMPylated proteins being falsely identified as ADP-ribosylated proteins.

### Addition of olaparib during lysis and avoiding boiling of samples are essential for the detection of endogenous ADP-ribosylation states

After having determined their specificity on in vitro substrates, we next determined which signals the antibodies detect in HEK293T lysates. We overexpressed almost all full-length GFP-tagged PARPs, and GFP alone, and first confirmed the expression of these proteins (Fig 3A). PARP2 was not included because of its expected redundancy with PARP1, and for PARP14, we were not able to generate a full-length construct containing an intact N-terminus. The overexpression of these enzymes led to the degradation of some enzymes, apparent from the smaller products visible in the anti-GFP blot. We nevertheless analysed the MARylation signal using Reagent V. We used this reagent to optimise experimental conditions, as it gives robust signals on the in vitro substrates and is commercially available in large amounts. We observed a relatively equal signal in most lanes, including the GFP-transfected control lane (Fig 3B). PARP1 is highly abundant and activated by damaged DNA, which is available during our RIPA-based lysis. To avoid false PAR/MAR signals from PARylation introduced by PARP1 during cell lysis, we performed all subsequent experiments with the addition of the PARP inhibitor olaparib to the lysis buffer. When preventing PARylation during lysis, the pattern detected by the antibody changes dramatically (Fig 3C): no activity can be seen for most of the poly-ARTs, with the exception of TNKS2, which could be partially caused by omitting a PARG inhibitor during lysis. A signal is present for most of the mono-ARTs. The overexpression of PARP6, PARP 7, PARP8, PARP10, and PARP15 leads to the modification of diverse proteins, whereas for PARP11 and PARP12, only a few substrates were detected. As it was shown that the inhibition of the proteasomal degradation system may lead to an increase in PARP7 MARylation (Lu et al, 2019), we tested whether the inhibition of the proteasome leads to an increase in the signal in our experiments. For most enzymes, we observed only slight differences in the MARylation signal after proteasomal inhibition; however, for some PARPs the signal appeared stronger upon proteasomal inhibition (Fig 3C). This may imply that MARylation has a high turnover and accumulates upon proteasomal inhibition. For other proteins, like PARP12, only distinct bands, which may correspond to the overexpressed proteins themselves, are increased upon MG132 treatment. It is possible that the MARylation signal that is present upon the overexpression of individual PARPs is not derived from modification introduced by this PARP, but from another PARP that became activated upon the introduction of the exogenous enzyme. There are examples of PARPs modifying one another, as PARP7, for example, is capable of modifying PARP13 (Rodriguez et al, 2021). When lysing cells under basal conditions, only a low signal for endogenous MARylation is present. This signal is not increased upon

transfection of, for example, a GFP-encoding plasmid, indicating that it is not the transfection stress that activates the endogenous ARTs. Future studies using inactive mutants and/or chemical genetics, as reviewed here (Rodriguez et al, 2021), are needed to further decipher which PARPs can potentially modify each other. We also noted that Western blots were on occasion not reproducible when performed only a few days apart using the same lysates, which prompted us to test the effect of different sample preparation conditions on Western blot outcome. For these analyses, we reused the samples from Fig 3C and analysed the enzymes that rendered the highest MARylation signal. When samples were frozen, boiled for 5 min in SDS sample buffer, and analysed on gel again, the signal-to-noise ratio was worse, implying an increase in unspecific modification or a loss of specific modification upon freezing and boiling of the samples (Fig 3D and E). When loading the same frozen samples without boiling, a few specific bands seem to be preserved, which were lost upon boiling, for example, in the PARP11 and PARP12 samples (Fig 3D); however, also here the signal-to-noise ratio is worse than it was in the initial analysis (Fig 3C). It is in theory possible that boiling influences the entering of proteins into the gel and therefore leads to an apparent loss of modification. Previous work has also demonstrated the thermolability of the cysteine ADP-ribosylation of GAP-43, as ~50% of the modification is lost upon incubation for 10 min at 95°C (Philibert & Zwiers, 1995). We used the same peptides and the automodified PARP10 catalytic domain, which were slot-blotted, untreated, or heated at different temperatures to test whether the modification is thermolabile (Figs 3F and S4). As samples are blotted directly onto the membrane using a vacuum without prior electrophoresis, slot blotting excludes the possibility that heating interferes with electrophoresis and therefore detectable signal. We observe that the modification of the peptides and PARP10 decreases with incubation time and temperature. At 95°C, the modification is lost rapidly from all peptides and from PARP10 itself, whereas at 60°C, the modification not only appears more stable but also diminishes with longer incubation times. It is possible that also on other amino acids, the modification is thermolabile. In subsequent experiments, we always analysed fresh lysates, prepared with olaparib and only briefly heated at 60°C. We also tested the influence of pH on stability and found that the pH of 7.4 of our lysis buffer is suitable for all amino acid linkages tested (Fig S5).

### Not all reagents perform equally well on endogenous ADP-ribosylation

Having established the conditions best suited for lysis, we next analysed ARTs, which modify different targets. TNKS2 was included as poly-ART, PARP7 as a cysteine-modifying enzyme, murine ART2.2 for arginine, PARP10 for its postulated major modification of glutamate, and PARP6 as an example of ART for which the specificity has not been determined yet. Reagents IV and V show robust modification signals for most enzymes (Fig 4A and B). In general, Reagents IV–VI detect the arginine modification introduced by mART2.2 well (Fig 4A–C). In contrast, a TNKS2 signal is present when using the anti-PAR Reagent VII as expected (Fig 4D). Reagent II detects most substrates, both polymers from TNKS2 and MARylation from the different enzymes (Fig 4E). Reagent I detects MARylation

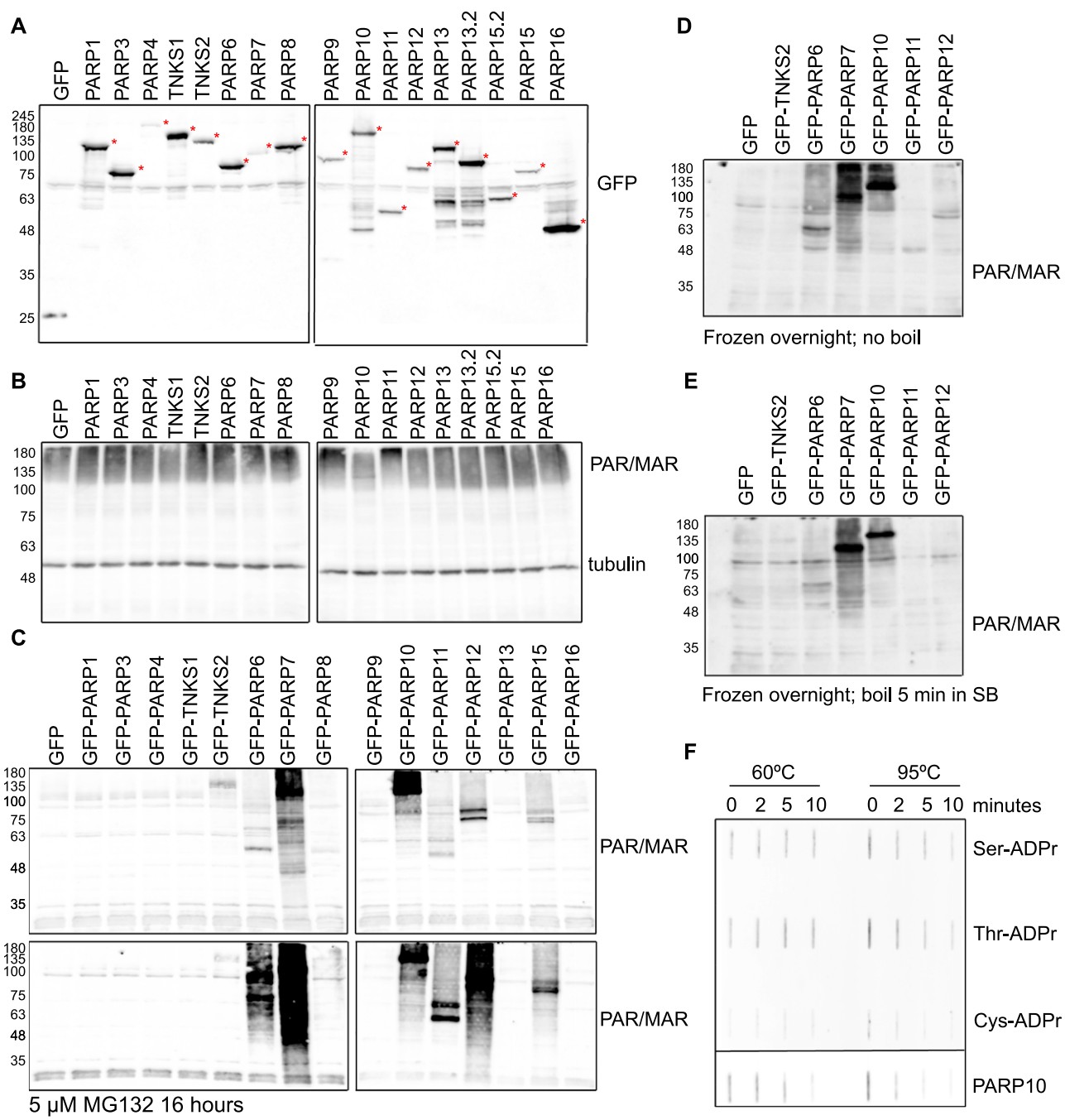

**Figure 3. Optimised lysis conditions are required to prevent ADP-ribosylation from occurring during lysis or degradation during sample preparation.**
**(A)** HEK293T cells were transfected with indicated GFP-PARPs, lysed in RIPA buffer, and analysed using Western blotting with a GFP antibody. **(B)** Same lysates as in (A), but analysed with ADPr antibody Reagent V. Subequently the blot was detected using a tubulin antibody. **(C)** HEK293T cells were transfected with indicated GFP-PARPs and lysed in RIPA buffer supplemented with olaparib or in addition treated with proteasomal inhibitor MG132 before lysis. Western blots were analysed with ADPr antibody Reagent V. **(D, E)** Untreated lysates from (C) were frozen at −20°C and heated at (D) 60°C or (E) 95°C before loading the SDS–PAGE. Resulting Western blots were analysed with ADPr antibody Reagent V. **(F)** ADP-ribosylated peptides and automodified PARP10 catalytic domain were slot-blotted, untreated, or heated at 60°C and 95°C for 2, 5, or 10 min. The blot was analysed using an ADPr antibody Reagent V. The PARP10 blot was exposed shorter because of the stronger signal; different exposures are provided as source data. Source data are available for this figure.

introduced by PARP6, PARP7, and PARP10 (Fig 4F). The weakest but perhaps most specific signal is visible when the blots are probed with Reagent III (Fig 4G), as here only PARP7 and PARP10 lead to clear signals. These signals are absent with the binding-deficient

mutant Reagent IIIm (Fig 4H). The expression of the GFP constructs was determined using an anti-GFP Western blot (Fig 4I). Reagent I, which is based on the human macrodomains from PARP14, in addition detects arginine. The greatest variability is visible for

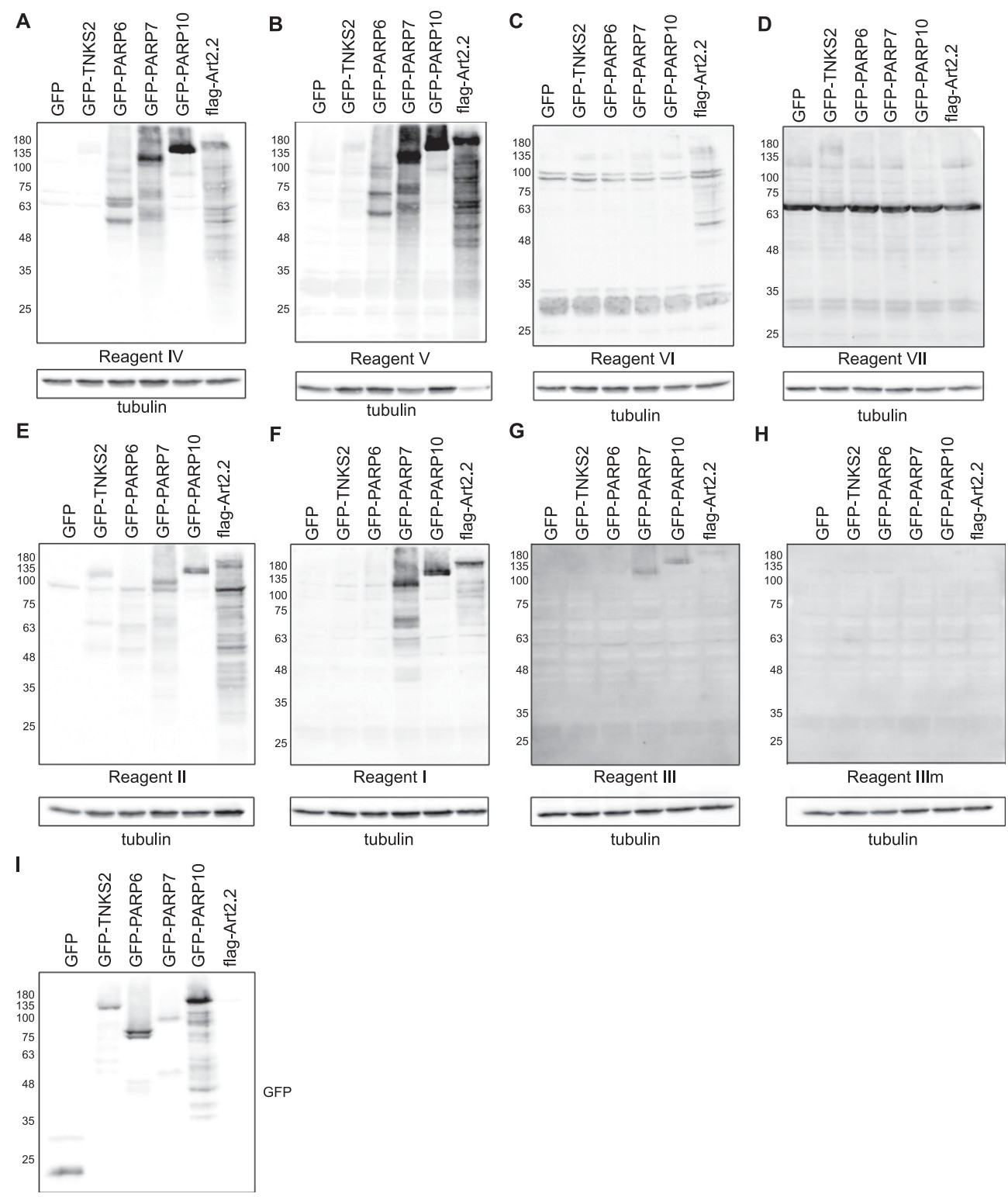

**Figure 4.  Anti-ADP-ribose detection reagents have different specificities for substrates modified in cells.**
**(A, B, C, D, E, F, G, H)** HEK293T cells were transfected with the indicated GFP-tagged PARP constructs, GFP as control or flag-tagged murine ART2.2. 24 h after transfection, cells were lysed in RIPA buffer supplemented with olaparib and analysed using SDS–PAGE. The Western blot was detected with the indicated reagents. **(I)** Western blot showing the transfection levels of the GFP-transferases.
Source data are available for this figure.

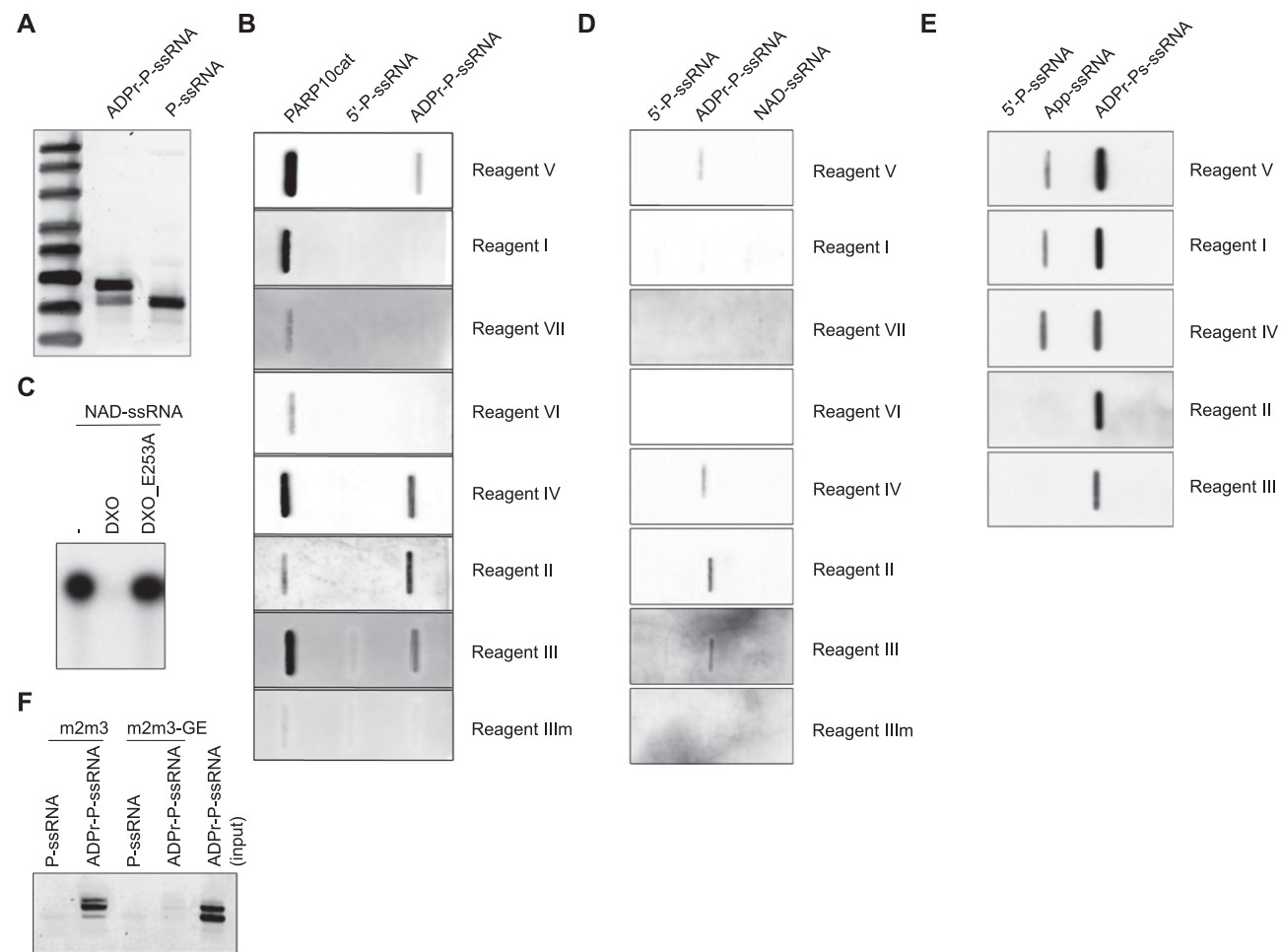

**Figure 5. Subset of the ADP-ribosylation reagents can be used to detect and enrich MARylated ssRNA but not NAD⁺-capped RNA.**
**(A)** Phosphorylated, single-stranded RNA (P-ssRNA) was ADP-ribosylated with the catalytic domain of PARP10 ranging from amino acids N818–T1025 (PARP10cat), resolved on urea–PAGE, and analysed using SYBR Gold staining. **(B)** Slot blot was performed with automodified PARP10cat, 5′P-ssRNA, or ADPr-ssRNA. The MARylated ssRNA was treated with proteinase K and purified before slot blotting. The blot was analysed using the indicated detection reagents. **(C)** Product of an in vitro transcription reaction with $^{32}$P-labelled NAD⁺ was analysed using urea–PAGE and autoradiography. DXO specifically removes the NAD⁺-cap and degrades the RNA oligo, whereas an inactive mutant is unable to remove the NAD⁺-cap. **(D)** As in (B), but now with NAD⁺-capped RNAs. NAD⁺-capped RNAs were generated during an in vitro transcription reaction, treated with proteinase K, and purified before analysis by slot blotting. **(E)** Adenylylated RNAs were slot-blotted and analysed using the indicated reagents. ADP-ribosylated RNA was slot-blotted as a positive control, and phosphorylated RNA, as a negative control. **(F)** 5′P-ssRNA or ADPr-ssRNA generated as in (A) was incubated with magnetic agarose beads coated with the indicated mPARP14-macro2/3 constructs. After extensive washing, samples were eluted from the beads and analysed on urea–PAGE.

PARP6: some reagents detect no signal at all, such as reagents I and VI, where others detect robust MARylation. This highlights the necessity of ongoing reagent evaluation, as it appears that the suitability of a reagent is dependent on the enzyme of interest.

### Only some reagents are suitable to detect RNA ADP-ribosylation

As recent publications have shown that also nucleic acids can be MARylated, we next determined whether these different reagents can be used to detect the modified nucleic acids. We generated MARylated RNA oligonucleotides and confirmed their modification using denaturing urea–PAGE and SYBR Gold staining to visualise the nucleic acids in gel (Fig 5A). We purified the different RNA species and slot-blotted them alongside automodified PARP10 as control (Fig 5B). Several reagents can be used to detect MARylated RNA:

both the antibody Reagents IV and V and the macrodomain-based Reagents II and III detect the modification. We performed a similar experiment with NAD⁺-capped RNA, which we produced as described before (Jiao et al, 2017). To verify capping, we generated RNA with a radiolabelled NAD⁺-cap and subjected this to DXO treatment, which degrades specifically NAD⁺-capped RNAs (Jiao et al, 2017) (Fig 5C). None of the antibodies tested can be used to detect the NAD⁺-cap (Fig 5D), which might imply that the ribose through which the NAD⁺ is attached to the RNA is part of the epitope and not available after linking to RNA. Adenylylation closely resembles ADP-ribosylation and may be recognised by these reagents. We generated 5′-adenylylated ssRNA and slot-blotted it alongside ADP-ribosylated RNA. Reagents I, IV, and V can detect the modification, whereas the macrodomain-based Reagents II and III are not able to detect it (Fig 5E). This further confirms that the reagents

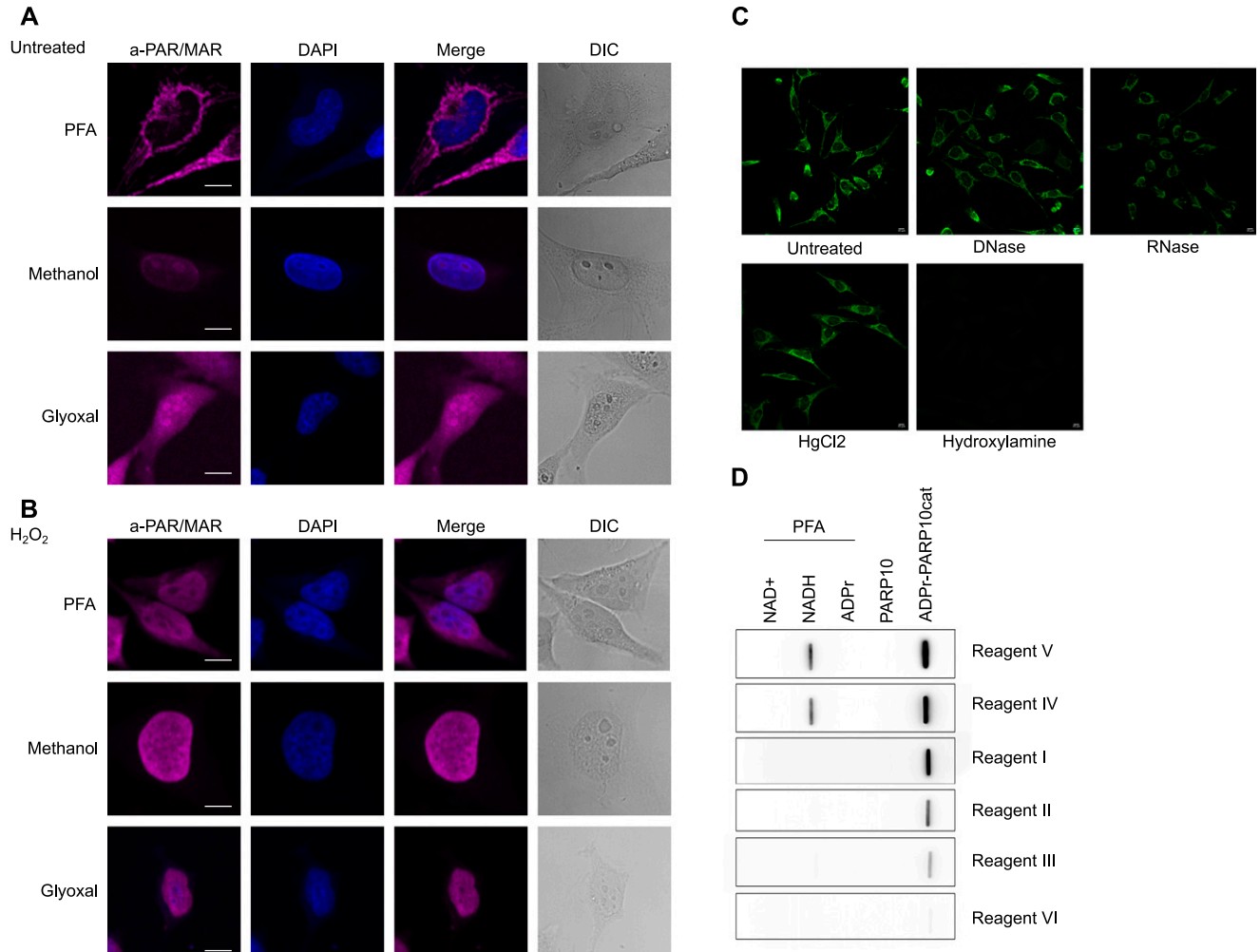

**Figure 6. ADP-ribosylation detection reagents stain different structures depending on the fixation method used.**
**(A)** HeLa cells were seeded onto glass coverslips and fixed using PFA, ice-cold methanol, or glyoxal. Reagent V was used to stain the cells, and DAPI was applied to stain the nuclei. **(B)** As in (A), but before fixation cells were treated with 0.5 mM H$_2$O$_2$ for 10 min. All images were taken using a confocal microscope with identical laser intensity and settings across all samples. **(C)** HeLa cells were seeded as in (A) and fixed using PFA. Fixed samples were treated with either DNase or RNase, HgCl$_2$, or hydroxylamine. ADP-ribosylation was visualised using Reagent V, and images were taken using confocal microscopy at the same settings for each sample. **(D)** Indicated metabolites were cross-linked to BSA using PFA and slot-blotted. Blots were incubated with the indicated antibodies and detected using chemiluminescence. Scale bars in (A, B, C) represent 10 µM.

recognise significantly different epitopes. Lastly, we enriched in vitro ADP-ribosylated ssRNA using the GFP-mPARP14-m2m3-Fc wild-type or GE mutant. mPARP14-m2m3 efficiently binds the modified ssRNA, but not the unmodified RNA (Fig 5F). The binding mutant does not interact with either modified or non-modified RNA. In this pull-down assay, an additional band is enriched with m2m3 wild type, which could imply that PARP10cat can generate an oligomer, which the module preferentially binds, or that it modifies multiple sites on each RNA, also leading to enhanced precipitation. In addition to their usage to study MARylated proteins, a subset of the available antibodies and detection reagents can thus be used to start studying the presence and function of nucleic acid ADP-ribosylation in cells. The macrodomain-based reagents are ideal to identify the modified RNAs in cells, as the binding mutants can be used to monitor non-specific binding to both module and column. These data also highlight that their

epitopes are diverse, as not all function equally well on nucleic acid substrates.

## Immunofluorescence staining of ADP-ribosylation is highly dependent on the fixation method

Lastly, we tested the reagents' suitability for immunofluorescence staining and confocal microscopy. We fixed cells using cross-linking agents PFA and glyoxal, and the dehydrating agent methanol. The CST antibody, Reagent V, gives the strongest signal: if identical laser strengths are used for the other reagents, the signals are very low (Figs 6A and S6). The relatively weak staining obtained with Reagent II, based on Af1521, could be caused by its catalytic activity during our procedures at room temperature, which should be restricted during our incubation for Western blot at 4°C. Using PFA as a fixative, we can confirm the cytoplasmic staining reported before

with Reagent V (Fig 6A) (Hopp & Hottiger, 2021a; Hopp et al, 2021). Glyoxal fixation in general resembles the methanol-fixed samples and leads to low staining intensity with most of the reagents, but gives rise to nucleolar staining with the CST antibody (Fig 6A). Upon $H_2O_2$ treatment, the ADP-ribosylation signal becomes strongest in the nucleus (Fig 6B). The discrepancy between cytoplasmic stainings in untreated cells fixed with PFA can have several possible explanations: either the respective epitopes are not exposed after fixing with methanol, or something is stained that is washed out by methanol but cross-linked by PFA, such as small metabolites. We expected similar staining patterns between PFA and glyoxal, as both are cross-linking agents. One key difference, however, is that glyoxal, in contrast to PFA, does not cross-link RNA. In theory, it is possible that the strong cytoplasmic signal observed with a number of the reagents is mitochondrial RNA or a metabolite. Both antibodies that efficiently recognise slot-blotted MARylated ssRNA also stain cytoplasmic structures after PFA fixation but not after glyoxal fixation. We incubated the slides with either RNase or DNase to remove signals derived from potentially MARylated RNA or DNA, neutral hydroxylamine to reverse the modification of acidic residues, or mercury chloride to reverse the modification of cysteines. All treatments lead to some reduction in the signal; however, the strongest change is visible in the hydroxylamine-treated samples, which may indicate that glutamate-linked ADP-ribosylation was detected (Fig 6C). Hydroxylamine is also able to reverse PFA cross-links and may thus have released the antigen. Recent data suggest that certain antibodies detect NADH (Hottiger M, personal communication), which we tested by cross-linking $NAD^+$, NADH, and ADPr to BSA using PFA and slot blotting the cross-linked metabolites. In these experiments, multiple reagents detect cross-linked NADH although not equally efficient, as the exposure time of the blots differs (Fig 6D). The mitochondrial staining is thus most likely due to cross-linked NADH instead of ADP-ribosylated proteins. $NAD^+$ is probably not efficiently cross-linked under these conditions and therefore not detected. This would agree with the low level of ADP-ribosylated proteins detected in whole-cell lysates.

## Discussion

In this work, we set out to compare the affinities of the different reagents available to detect MARylation. After preprinting this study, we obtained two additional antibodies, which we tested on a number of in vitro substrates (Fig S7), to provide an overview of the currently available reagents as complete as possible. The reagents have different strengths and weaknesses (Table 2): many also recognise PAR, making them less suitable to study exclusively MARylation. The macrodomain-based reagents show higher specificity towards MARylated substrates, but in general bind weaker to the modified proteins with the exception of eAF1521, which is very efficient. A major advantage of the macrodomain-based reagents is their availability. They can be generated any time, thus avoiding the risk that reagents will become unavailable as was the case with Reagent VII (anti-PAR antibody from Trevigen), which is not available anymore. A further advantage is the possibility to create ADPr binding mutants, which can be used as a negative

control to eliminate the proteins that bind unspecifically. Despite weaker signals in Western blots, these macrodomains and their mutants may prove to be very useful to immunoprecipitate MARylated proteins and especially RNAs with high specificity.

As the concentration of the reagents provided by other researchers was partially unknown, we cannot determine their absolute affinities, but merely compare which reagent is best suited for the respective substrates. It should also be kept in mind that the production methods are different, ranging from purification from sera for polyclonal antibodies and hybridoma culture for monoclonal antibodies, to the supernatant of cultured cells for eAf1521 to *Escherichia coli* for mPARP14-m2m3-Fc. These differences could potentially influence the outcomes as well and may require further optimisation, especially for mPARP14-m2m3-Fc, where it is feasible to reduce the background noise by further optimisation of purification and storage conditions. Not only is the use of the correct detection reagent for specific substrates important, but we also illustrated how sample preparation can influence the apparent ADP-ribosylation signal. Routine boiling of Western blot samples may have detrimental effects and lead to false-negative results. The inhibition of PARP1 is necessary to prevent the induction of poly(ADP-ribosyl)ation during lysis, which may also be true for other ARTD family members and the reverse hydrolase reaction. Furthermore, ART and hydrolase inhibitors should be added during lysis or even to the cell cultures, if available. Similarly, the fixative chosen for immunofluorescence approaches influences the structures stained by the different reagents. Sample processing can thus hugely influence the experimental outcome and ought to be both considered and documented carefully.

An interesting question arising from the immunofluorescence images is why the mitochondria are stained only with some reagents and PFA fixation. Methanol fixation leads to the most consistent results: virtually, no signal is present in untreated cells with $H_2O_2$ leading to a signal in the nucleus. This agrees with the Western blots, where endogenous MARylation is very low when untreated cells are lysed and analysed. It is possible that during cell lysis, when mitochondrial contents are released, a highly active hydrolase reverses the modification of the mitochondrial proteins detected using IF. Alternatively, nucleic acids or metabolites are the source of mitochondrial ADP-ribosylation and not proteins. The fact that the antibodies that give rise to cytosolic staining in immunofluorescence also detect NADH strongly suggests that metabolites rather than proteins are stained.

The observed lack of MARylation in the cytosol is unexpected as many of the mono-ARTs are expressed in the cytosol (Vyas et al, 2013; Challa et al, 2021; Hopp & Hottiger, 2021b). The question remains which stimuli activate these enzymes in cells: Is there a generic signal that can activate them all, analogous to the caspases being activated in a cascade during regulated cell death? Or are there specific triggers that activate specific ARTs, such as possibly viral infection activating the interferon-responsive ARTs, high levels of unfolded proteins activating PARP16, or translational shutdown activating the stress granule-associated ARTs. The alternative explanation for the low MARylation signal is that the corresponding hydrolases are highly active and prevent visualisation of the MARylation because of high turnover rates. Increased knowledge about the hydrolases, and specific inhibitors or knockout cells, is required to distinguish between these two possible explanations for the lack of detectable endogenous MARylation. The availability

**Table 2.** Summary of the ADP-ribose detection reagent properties.

| | PAR/MAR detection | Western blot | ADPr RNA | IF | Caution |
|---|---|---|---|---|---|
| I Anti-MAR reagent | Suitable for MARylation | Good | Not suitable | Suitable with all fixatives | May depend on the backbone |
| II eAf1521-Fc | No distinction MAR/PARylation | Good | Suitable | Suitable with all fixatives | Retains catalytic activity |
| III mPARP14-macro2/3-Fc | Suitable for MARylation | Weak signal | Suitable | Suitable with all fixatives | May depend on the backbone |
| IV Anti-MAR antibody | No distinction MAR/PARylation | Good | Suitable | Yes, when fixed with MeOH or glyoxal | Cross-reacts with AMPylation and NADH |
| V Anti-PAR/MAR (E6F6A) | No distinction MAR/PARylation | Good | Suitable | Yes, when fixed with MeOH or glyoxal | Cross-reacts with NADH |
| VI Pan-ADPr | No distinction MAR/PARylation | Weak signal | Not suitable | Suitable with all fixatives | — |
| VII Anti-PAR antibody | Suitable for PARylation | Suitable for PARylation but high background | Not suitable | n.d. | Only PARylation |
| Anti-ADPr HCA354 | Suitable for MARylation | Good | Suitable | n.d. | Cross-reacts with AMPylation and NADH |
| Anti-ADPr HCA355 | Suitable for MARylation | Good | Not suitable | n.d. | — |

of several reagents capable of detecting MARylation on in vitro modified proteins, in cell lysates and in immunofluorescence, will enable future work studying these regulatory mechanisms. Likewise, some of the available reagents are suitable for the detection of MARylated RNA. It is not clear to which extent the in vitro assays represent the activity of the ARTs in cells: the sheer amounts of enzymes and substrates being brought together may induce the artificial modification of exposed residues. Measuring in-cell ADP-ribosylation will likely provide much more accurate information than in vitro assays.

One of the major outstanding questions in the field is thus to identify the signals that trigger endogenous ART activity in cells, and the extent wherein hydrolases reverse the modification in cells and during lysis. This work highlights the importance of careful sample preparation to avoid both loss and artificial gain of MARylation signal and provides a comparison of the currently available reagents, which will allow researchers to make an informed decision as to which reagent to use for their specific purposes.

# Materials and Methods

### Plasmids

Many plasmids used to express proteins were gifts from other laboratories, as indicated below (Table 3). For the expression of GFP-ARTs in mammalian cells, we amplified the different genes using appropriate primers for the full-length gene products from HeLa cDNA where possible or used gBlocks from IDT. These were transferred into the Gateway system. All ARTs harbour the GFP-tag on the N-terminus in these constructs. Full plasmid sequences are available with the plasmids on Addgene. The generation of pDONR-mPARP14-m2m3 constructs was described before (Forst et al, 2013).

These were transferred using the Gateway system to a Gateway-compatible pEGFP plasmid or amplified with appropriate primers for restriction cloning into pET28a.

### Protein purification

Most of the ARTs used in this study were purified from bacterial expression systems with N-terminal His- or GST-tags, with the exception of PARP1, which was produced as GFP-fusion protein from HEK293T cells as described below. As we found that many of the proteins are toxic to the bacteria, we optimised expression conditions and bacterial strains for each protein separately, with the optimal conditions summarised below (Table 4).

Bacterial cultures were spun down at 6,000$g$ for 15 min at 4°C, followed by resuspension in lysis buffer. Cell suspensions were sonicated on ice using Digital Sonifier 250 Cell Disruptor (Branson). Depending on the expressed protein construct and the pellet size, sonication varied in a range of 2–5 min at 15–20%, with 30 s on and 40 s off. The cell lysates were centrifuged at 45,000$g$ for 45 min at 4°C, to remove cell debris. The supernatant was used for affinity purification, using either glutathione–Sepharose 4G beads (GE Healthcare) or TALON Metal Affinity Resin (Takara), for GST-tagged or His-tagged constructs, respectively, followed by dialysis. The composition of lysis, wash, elution, and dialysis buffers was adjusted according to each protein's tag, properties, and isoelectric point, and is summarised in Table 5.

### Mammalian cell culture and transfection

All cells were cultured in DMEM with pyruvate, 4.5 g/l glucose (Gibco), and 10% heat-inactivated foetal calf serum (Gibco) in a humidified atmosphere with 5% $CO_2$. Cells are routinely tested for mycoplasma contamination and confirmed negative at the moment

**Table 3. Plasmids used in this study.**

| Plasmid name | Vector | Insert | Ref. | Addgene |
|---|---|---|---|---|
| pNH-TrxT-MACROD1_77-325 | pNH-TrxT | MACROD1_77-325 | Chen et al (2011) | n/a |
| pDEST17-MACROD2 | pDEST17 | MACROD2 | Rosenthal et al (2013) | 172593 |
| pDEST17-TARG1 | pDEST17 | TARG1 | Butepage et al (2018b) | 172594 |
| pNH-TrxT_PARGcat_448-976 | pNH-TrxT | PARGcat_448-976 | Wazir et al (2021) | n/a |
| pDEST17-DXO | pDEST17 | DXO | This study | 177993 |
| pDEST17-DXO_E253A | pDEST17 | DXO_E253A | This study | 177,994 |
| pNIC-Bsa4_ARH1 | pNIC-Bsa4 | ARH1 | Wazir et al (2021) | n/a |
| pDEST17-ARH3 | pDEST17 | ARH3 | This study | 183051 |
| pGST-PARP10_818-1025 | pGST | PARP10_818-1025 | Kleine et al (2008) | n/a |
| pASK60-mART2.2-FlagHis6x | pASK60 | Murine ART2.2 | Mueller-Dieckmann et al (2002) | n/a |
| pET15b-rPtxS1 | pET15b | rPtxS1 | Ashok et al (2020) | 173076 |
| pGEX-C3 bot | pGEX | *Clostridium botulinum* C3 toxin | Pautsch et al (2005) | n/a |
| pET28a-mPARP14-macro2/3-Fc | pET28a | Murine PARP14-macro2/3 | This study | |
| pET28a-mPARP14-macro2/3-GE-Fc | pET28a | Murine PARP14-macro2/3-GE | This study | |
| pEGFP-mPARP14-macro2/3-Fc | pEGFP-C1 | Murine PARP14-macro2/3 | This study | |
| pEGFP-mPARP14-macro2/3-GE-Fc | pEGFP-C1 | Murine PARP14-macro2/3-GE | This study | |
| pME_CD8L_FLAG-ART2.2GPI | pME | Murine ART2.2 | Koch-Nolte et al (1999) | n/a |
| pEGFP-PARP# | pEGFP | PARP1, PARP4, TNKS1, TNKS2, PARP8, PARP9, PARP13, PARP16 | Vyas et al (2013) | n/a |
| pcDNA5-FRT/TO-N-mEGFP-PARP3 | pcDNA5-FRT/TO-N-mEGFP | PARP3 | This study | 178007 |
| pcDNA5-FRT/TO-N-mEGFP-PARP6 | pcDNA5-FRT/TO- N-mEGFP | PARP6 | This study | 178005 |
| pcDNA5-FRT/TO-N-mEGFP-PARP7 | pcDNA5-FRT/TO- N-mEGFP | PARP7 | This study | 178004 |
| pcDNA5-FRT/TO-N-mEGFP-PARP10 | pcDNA5-FRT/TO- N-mEGFP | PARP10 | This study | 177997 |
| pcDNA5-FRT/TO-N-mEGFP-PARP11 | pcDNA5-FRT/TO- N-mEGFP | PARP11 | This study | 177998 |
| pcDNA5-FRT/TO-N-mEGFP-PARP12 | pcDNA5-FRT/TO- N-mEGFP | PARP12 | This study | 177999 |
| pcDNA5-FRT/TO-N-mEGFP-PARP15 | pcDNA5-FRT/TO- N-mEGFP | PARP15 | This study | 178000 |

of experiments. HEK293T cells were transfected using the calcium phosphate method as described in a step-by-step protocol available online (Feijs, 2021b). Briefly, cells were seeded onto six-well plates with $3 \times 10^5$ cells per well and transfected next day with 4 $\mu$g DNA per well. ~6 h after transfection, cells were washed and fresh full DMEM was added. 24–48 h after transfection, cells were lysed.

### Immunoprecipitation

GFP-PARP1 for enzymatic assays was immunoprecipitated from HEK293 cells. The cells were transfected with a plasmid encoding for GFP-PARP1 in 10-cm dishes and were further processed 24–48 h after transfection. Cells were washed in warm DMEM without FCS and lysed in CoIP buffer (10 mM Hepes, pH 7.5, 50 mM NaCl, 30 mM sodium $Na_4P_2O_7$, 50 mM NaF, 0.2% Triton X-100, and 10% glycerol) containing protease inhibitor cocktail (8340; Sigma-Aldrich) and Benzonase 1:10,000 (9025-65-4; Santa Cruz). Lysates were centrifuged at 4°C and the maximum speed for 15 min to remove insoluble material, followed by the incubation of the cleared supernatant with 10 $\mu$l prewashed GFP-coupled magnetic agarose (ChromoTek) per 10-cm dish. After half an hour under rotation in the cold room, beads were washed in CoIP buffer, followed by a final

**Table 4. Conditions for recombinant protein expression.**

| Protein | Plasmid | Bacterial strain | OD$_{600}$ at induction | IPTG [mM] | Time | Temp. | Source |
|---|---|---|---|---|---|---|---|
| MACROD1 | pNH-TrxT-MACROD1_77-325 | BL21-CodonPlus (DE3)-RIL | 0.6 | 0.4 | 16 h | 22°C | Chen et al (2011) |
| MACROD2 | pDEST17-MACROD2 | BL21-CodonPlus (DE3)-RIL | 0.6 | 0.4 | 16 h | 22°C | Rosenthal et al (2013) |
| TARG1 | pDEST17-TARG1 | BL21-CodonPlus (DE3)-RIL | 0.6 | 0.4 | 16 h | 22°C | Butepage et al (2018b) |
| PARGcat | pNH-TrxT_hPARG_cat_448-976 | Rosetta (DE3)pLysS | 0.6 | 0.4 | 16 h | 18°C | Wazir et al (2021) |
| ARH1 | pNIC-Bsa4_hARH1 | BL21-CodonPlus (DE3)-RIL | 0.6 | 1 | 16 h | 18°C | Wazir et al (2021) |
| ARH3 | pDEST17-ARH3 | BL21-CodonPlus (DE3)-RIL | 0.6 | 1 | 6 h | 16°C | Kleine et al (2008) |
| PARP10cat | pGST-PARP10_818-1025 | BL21-CodonPlus (DE3)-RIL | 0.6 | 1 | 16 h | 18°C | Kleine et al (2008) |
| mART2.2 | pASK60-ART2.2-FlagHis6x | Lemo21(DE3) *E. coli* | 0.6 | 1 | 16 h | 16°C | Mueller-Dieckmann et al (2002) |
| Pertussis toxin_cat | pET15b-rPtxS1 | BL21-CodonPlus (DE3)-RIL | 0.6 | 0.4 | 16 h | 18°C | Ashok et al (2020) |
| C3 botulinum | pGEX-C3-Cbot | Rosetta (DE3)pLysS | 0.6 | 0.4 | 16 h | 18°C | Pautsch et al (2005) |
| mPARP14-macro2/3-Fc | pET28-PARP14-macro2/3-Fc | Rosetta (DE3)pLysS | 0.6 | 0.4 | 16 h | 18°C | This study |
| mPARP14-macro2/3-GE-Fc | pET28-PARP14-macro2/3-Fc-GE | Rosetta (DE3)pLysS | 0.6 | 0.4 | 16 h | 18°C | This study |

wash in 1× PARP assay buffer. The material from one dish was split into four tubes for subsequent enzymatic reactions. Samples were processed immediately or stored in one-use aliquots at −20°C.

### Generation of cytosolic extracts

To generate substrate protein for some of the transferases and toxins, cytosolic extracts were generated. HEK293T cells were grown on 15-cm dishes until fully confluent and collected by trypsinisation. Cells were washed in hypotonic buffer (250 mM sucrose, 2 mM Hepes, and 0.1 mM EGTA, pH 7.4), followed by incubation for ~15 min on ice to allow swelling, as monitored using a microscope. Cells were then broken by 20 strokes with a Dounce pestle B, followed by centrifugation at 11,000$g$ to remove unbroken cells, nuclei, and mitochondria. The resulting extracts are devoid of transferase PARP1 and devoid of most hydrolases, which are either mitochondrial, nuclear, or expressed at low levels (Niere et al, 2008; Zaja et al, 2020).

### ADP-ribosylation and hydrolase assays

Protein ADP-ribosylation assays were routinely carried out at 37°C for 30 min unless indicated otherwise. Reactions were carried out in 30 $\mu$l containing 50 mM Tris–HCl, pH 7.5, 0.5 mM DTT, 0.1% Triton X-100, 5 mM MgCl$_2$, and 50 $\mu$M $\beta$-NAD$^+$ (Sigma-Aldrich) and 1 $\mu$Ci [$^{32}$P]-$\beta$-NAD$^+$ (Hartman Analytics). 15 $\mu$l HEK293T cytosolic extract was used for the toxin and mART2.2 reactions. The amounts of enzymes used varied depending on the activity of the enzyme

studied. Reactions were placed on ice, and where possible, transferase inhibitors were added before adding hydrolases. For PARP10, OUL35 was used at 2 $\mu$M; for PARP1, olaparib was added at 5 $\mu$M. For mART2.2 and the toxins, no inhibitors were available and hydrolases were added to the cooled reaction. Hydrolase reactions were incubated at 37°C for 30 min, stopped by adding 4× SDS sample buffer, heated for 10 min at 60°C, and run on SDS–PAGE. If protein coupled to a GFP-trap was analysed, we incubated the beads for 5 min at 95°C before loading on gel. Gels were dried, and incorporated radioactivity was analysed by exposure of the dried gel to X-ray film.

For RNA modification, 7 $\mu$M of ssRNA was incubated with 2 $\mu$M TRPT1 in ADPr buffer (20 mM Hepes–KOH, pH 7.6, 50 mM KCl, 5 mM MgCl$_2$, 1 mM DTT, 500 $\mu$M NAD, and 40U RNase inhibitor) at 37°C for 60 min while shaking. TRPT1 was digested by proteinase K treatment (20 U), for 20 min at RT, before purification using the Monarch RNA Cleanup Kit (T2030). Concentration was determined by spectrophotometric measurements (NanoDrop ND-1000 Spectrophotometer).

### Antibodies, Western blot, and slot blot

Cells were washed in warm DMEM without FCS before lysis to avoid unnecessary stress. Cell protein extractions were performed using RIPA buffer (150 mM NaCl, 1% Triton X-100, 0.5% sodium deoxycholate, 0.1% SDS, and 50 mM Tris–HCl [pH 8.0]) supplemented with protease inhibitor cocktail, Benzonase, and olaparib. Proteins were separated on 10–15% gels and blotted onto nitrocellulose membranes using a Bio-Rad TurboBlot apparatus using the high

**Table 5.  Protein purification conditions.**

| Protein | Lysis buffer | Wash I | Wash II | Elution buffer | Dialysis buffer |
|---|---|---|---|---|---|
| His-MACROD1 | 20 mM Hepes/NaOH, pH 8.0, 300 mM NaCl, 5 mM imidazole, 0.1% NP-40, 10% glycerol, 1 mM TCEP, PIC, Benzonase | 20 mM Hepes/NaOH, pH 8.0, 300 mM NaCl, 10 mM imidazole, 0.1% NP-40 | | 20 mM Hepes/NaOH, pH 8.0, 300 mM NaCl, 300 mM imidazole | 30 mM Hepes/NaOH, pH 8.5, 200 mM NaCl, 10% glycerol, 1 mM TCEP |
| His-MACROD2 | | | | | 20 mM Tris–HCl, pH 6.4, 200 mM NaCl, 10% glycerol, 1 mM TCEP |
| His-TARG1 | | | | | 20 mM Hepes/NaOH, pH 7.5, 200 mM NaCl, 10% glycerol, 1 mM TCEP |
| His-PARGcat | | | | 20 mM Hepes/NaOH, pH 8.0, 300 mM NaCl, 300 mM imidazole, 1 mM TCEP | 25 mM Hepes/NaOH, pH 8.0, 300 mM NaCl, 10% glycerol, 1 mM TCEP |
| His-ARH1 | 50 mM Tris–HCl, pH 8.0, 300 mM NaCl, 5 mM MgCl, 1 mM TCEP, 10 mM imidazole, PIC, Benzonase | 50 mM Tris–HCl, pH 8.0, 150 mM NaCl, 5 mM MgCl, 1 mM TCEP, 10 mM imidazole | | 50 mM Tris–HCl, pH 8.0, 150 mM NaCl, 5 mM MgCl, 1 mM TCEP, 300 mM imidazole | 50 mM Tris–HCl, pH 7.0, 150 mM NaCl, 10% glycerol, 1 mM TCEP |
| His-ARH3 | | | | | 50 mM Tris–HCl, pH 7.0, 150 mM NaCl, 10% glycerol, 1 mM TCEP |
| GST-PARP10_cat | 20 mM Tris–HCl, pH 8.0, 150 mM NaCl, 1 mM EDTA, 5 mM DTT, PIC, Benzonase | 50 mM Tris–HCl, pH 8.0, 150 mM NaCl | | 50 mM Tris–HCl, pH 8.0, 150 mM NaCl, 20 mM glutathione | 25 mM Tris–HCl, pH 7.5, 150 mM NaCl, 10% glycerol, 1 mM TCEP |
| His-mART2.2 | 20 mM Tris–HCl, pH 7.5, 300 mM NaCl, 1 mM TCEP, 500 µg/ml lysozyme, PIC, Benzonase | 50 mM Tris–HCl, pH 7.5, 150 mM NaCl, 1 mM TCEP, 10 mM imidazole | | 50 mM Tris–HCl, pH 7.5, 150 mM NaCl, 10% glycerol, 250 mM imidazole | 50 mM Tris–HCl, pH 7.5, 150 mM NaCl, 10% glycerol, 0.5 mM TCEP |
| His-Pertussis toxin_cat | 25 mM Tris–HCl, pH 8.0, 500 mM NaCl, 10% glycerol, 1 mM TCEP, 10 mM imidazole, PIC, Benzonase | 50 mM Tris–HCl, pH 8.0, 500 mM NaCl, 1 mM TCEP, 10 mM imidazole | 50 mM Tris–HCl, pH 8.0, 200 mM NaCl, 1 mM TCEP, 10 mM imidazole | 25 mM Tris–HCl, pH 8.0, 500 mM NaCl, 300 mM imidazole, 1 mM TCEP | 25 mM Hepes/NaOH, pH 8.0, 500 mM NaCl, 10% glycerol, 1 mM TCEP |
| His-mPARP14-m2m3-Fc | 20 mM Hepes/NaOH, pH 8.0, 300 mM NaCl, 10 mM Imidazole, 0.1% NP-40, 10% glycerol, 1 mM TCEP, PIC, Benzonase | 20 mM Hepes/NaOH, pH 8.0, 300 mM NaCl, 10 mM imidazole, 0.1% NP-40 | 20 mM Hepes/NaOH, pH 8.0, 150 mM NaCl, 10 mM imidazole | 20 mM Tris–HCl, pH 8.0, 300 mM NaCl, 300 mM imidazole, 1 mM TCEP | 25 mM Hepes/NaOH, pH 8.0, 300 mM NaCl, 10% glycerol, 1 mM TCEP |
| His-mPARP14-m2m3-GE-Fc | | | | | |
| His-DXO | 25 mM Hepes/NaOH, pH 7.0, 300 mM NaCl, 5 mM MgCl, 1 mM TCEP, 10 mM imidazole, 0.1% NP-40, PIC, Benzonase | 25 mM Hepes/NaOH, pH 7.0, 150 mM NaCl, 5 mM MgCl, 1 mM TCEP, 10 mM imidazole | 25 mM Hepes/NaOH, pH 7.0, 200 mM NaCl, 5 mM MgCl, 1 mM TCEP, 300 mM imidazole | | 25 mM Hepes/NaOH, pH 7.0, 150 mM NaCl, 10% glycerol, 1 mM TCEP |

molecular weight 10-min program. A step-by-step protocol for our Western blotting procedure is available online (Feijs, 2021a). Membranes were blocked with 5% non-fat milk in TBST for 30–60 min at RT, primary antibodies were diluted in TBST as indicated below and incubated overnight at 4°C, and secondary antibodies were diluted 1:5,000 in 5% non-fat milk in TBST and incubated for 30–60 min at RT. Some of the antibodies could be stored and reused multiple times as indicated in Table 2. Multiple wash steps were performed in between and after the antibody incubation with TBST at RT for at least 5 min. Chemiluminescent signals were detected by either exposure to film or using the Azure 600 equipment. Additional antibodies used were as follows: anti-GFP (1:2,500; 600-101-215; Rockland), anti-HSP60 (1:2,500; 12165; Cell Signaling Technology), anti-PARP1 (1:1,000; 1 835 238; Roche), and anti-tubulin (1:5,000; B-5-1-2; Santa Cruz). For slot blotting, 2 $\mu$M of each peptide was slot-blotted using a PR648 Slot Blot Blotting Manifold (Hoefer) onto a nitrocellulose membrane. Subsequently, the membrane was processed identical to the described processing of Western blots and detected using exposure to film.

### RNA and peptide slot blot

RNA was blotted on Hybond-N membrane (Amersham) using a PR648 Slot Blot Blotting Manifold (Hoefer). The membrane was activated in SSC buffer, pH 7.0 (150 mM NaCl and 15 mM NaCit), for 5 min. After assembling of the blotting sandwich, slots were flushed

**Table 6. Antibody dilutions.**

| # | Reagent | Source | Catalogue number | Dilution used for Western blot | Dilution used for microscopy | Isotype | Reusable? | Ref |
|---|---------|--------|------------------|-------------------------------|------------------------------|---------|-----------|-----|
| I | Anti-MAR | Millipore | MABE1076 | 1:1,000 | 1:500 | Rabbit IgG | No | Gibson et al (2017) |
| II | eAf1521-Fc | Michael Hottiger | n/a | 1:1,000 | 1:50 | Mouse IgG | Yes[a] | Nowak et al (2020) |
| III | mPARP14-macro2/3-Fc | Bernhard Lüscher | n/a | 1:1,000 | 1:50 | Mouse IgG | No | — |
| IIIm | mPARP14-macro2/3-GE-Fc | Bernhard Lüscher | n/a | 1:1,000 | 1:50 | Mouse IgG | No | — |
| IV | Anti-MAR | Michael Hottiger | n/a | 1:4,000 | 1:500 | Rabbit IgG | Yes[a] | Hopp et al (2021) |
| V | Anti-PAR/MAR (E6F6A) | Cell Signaling Technology | #83732 | 1:50,000 | 1:1,000 | Rabbit IgG | Yes[a] | Lu et al (2019) |
| VI | Pan-ADPr | Ivan Matic | 33,641 | 1:1,000 | 1:500 | Human | No | Bonfiglio et al (2020) |
| VII | Anti-PAR | Trevigen | 4336-BPC-100 | 1:1,000 | 1:100 | Rabbit IgG | Yes[a] | — |

[a]Stored at 4°C in TBST with 0.05% sodium azide.

with SSC buffer. 40 ng RNA in 200 $\mu$l SSC buffer per slot was applied to the membrane. The membrane was air-dried, and samples were cross-linked using UV light (120 mJ/cm$^2$). The membrane was blocked with 5% non-fat milk in PBST for 60 min at RT, primary antibodies were diluted in PBST as indicated and incubated overnight at 4°C, and secondary antibodies were diluted 1:5,000 in 2% non-fat milk in PBST and incubated for 30 min at RT. Multiple wash steps were performed both in between and after the antibody incubation with PBST for at least 5 min. Chemiluminescent signals were detected using the Azure 600 equipment. Peptides were blotted onto nitrocellulose activated in water, followed by blocking with 5% non-fat milk in PBST and antibody incubation. Antibody dilutions and wash steps were identical to the processing of Western blots. For slot blots with metabolites, 8 mM metabolites were cross-linked to 0.8 mM BSA using 4% PFA for 20 min at RT, followed by incubation with 100 $\mu$M Tris–HCL, pH 7.5, for 5 min at RT. 20 nM cross-linked metabolites were slot-blotted on a nitrocellulose membrane, which was blocked with 5% non-fat milk in PBST and incubated with primary antibodies overnight at the described dilutions.

### Confocal microscopy

U2OS or HeLa cells were seeded onto glass coverslips in 24-well plates. For $H_2O_2$ treatment, the medium was removed and replaced with warm PBS containing 1 mM $H_2O_2$ for 5 min. Cells were washed once in warm DMEM without FCS and fixed for 20 min with 4% PFA in PBS at room temperature. Alternatively, ice-cold methanol was added to the cells on coverslips after the removal of the growth medium, followed by 5-min incubation on ice and subsequent quick washing in PBS. For glyoxal fixation, glyoxal solution was added to the cells for 30 min on ice followed by 30 min at room temperature. After washing in PBS, the samples were quenched using 100 mM ammonium chloride and finally permeabilised using 0.1% Triton X-100. Regardless of the fixation method, subsequent blocking was done in PBS supplemented with 1% BSA in PBS with 0.1% Triton X-100 for 1 h at RT. Primary and secondary antibodies were applied for 1 h, with extensive washing in between. Primary antibody dilutions are indicated (Table 6). Secondary antibodies used were as follows: Alexa Fluor 594 anti-mouse, Alexa Fluor 594 anti-rabbit, and Alexa Fluor 633 anti-human (all from Thermo Fisher Scientific) used at dilutions 1:2,000 in PBS with 0.2% BSA. After extensive washing in PBS and demineralised water, coverslips were mounted on microscopy slides using Prolong Anti-Fade Diamond Mountant containing DAPI (Thermo Fisher Scientific). The samples were analysed with a Zeiss LSM 710 confocal laser scanning microscope equipped with an Axiocam (Zeiss) and a C-Apochromat 20× objective. Step-by-step immunofluorescence protocols are available online (Feijs, 2021c).

## Data Availability

Unprocessed blots are presented either in the article or in the Supplementary Data Files. Expression constructs generated in this study will be available from Addgene.

## Supplementary Information

## Acknowledgements

We thank the researchers in the ADP-ribosylation community who have put their expression constructs and detection reagents at our disposal to allow

thorough characterisation of these materials: eAf1521 and an ADPr antibody were provided by Michael O Hottiger; a pan-ADPr antibody was provided by Ivan Matic; the SpvB and mART2.2 bacterial and human expression plasmids were a gift from Fritz Koch-Nolte; truncated pertussis toxin, PARG, and ARH1 constructs were kindly provided by Lari Lehtiö; GFP-PARP expression constructs were provided by Paul Chang; purified recombinant HPF1 was provided by Patricia Korn; and AMPylated proteins were kindly provided by Aymelt Itzen and Dorothea Höpfner. We are grateful for helpful discussions with and suggestions from Michael O Hottiger, who pointed out cross-reactivity with NADH, and Michael Cohen and Daniel Bejan, who detected cross-reactivity with AMPylation. This work was supported by the Confocal Microscopy Facility, a Core Facility of the Interdisciplinary Center for Clinical Research (IZKF) Aachen within the Faculty of Medicine at RWTH Aachen University. Funding was provided by the START Program of the Medical Faculty of RWTH Aachen University to KLH Feijs (10/18) and R Žaja (13/20) and by the German Research Foundation DFG to B Lüscher (LU466/16-2) and KLH Feijs (FE1423/3-1).

## Author Contributions

L Weixler: investigation and writing—review and editing.
NJ Ikenga: investigation.
J Voorneveld: resources.
G Aydin: investigation.
TMHR Bolte: investigation.
J Momoh: investigation.
M Bütepage: resources.
A Golzmann: resources.
B Lüscher: resources and funding acquisition.
DV Filippov: resources.
R Žaja: conceptualisation, supervision, funding acquisition, investigation, and writing—review and editing.
KLH Feijs: conceptualisation, investigation, supervision, funding acquisition, visualisation, and writing—original draft, review, and editing.

## Conflict of Interest Statement

The authors declare that they have no conflict of interest.

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
