## [Reviewer comments · Life Science Alliance]

Life Science Alliance

Protein and RNA ADP-ribosylation detection is influenced by sample preparation and reagents used

Lisa Weixler, Nonso Ikenga, Jim Voorneveld, Gülcan Aydin, Timo Bolte, Jeffrey Momoh, Mareike Bütepage, Alexandra Golzmann, Bernhard Lüscher, Dmitri Filippov, Roko Zaja, and Karla Feijs

DOI: <https://doi.org/10.26508/lsa.202201455>

Corresponding author(s): Karla Feijs, Institute of Biochemistry and Molecular Biology, RWTH Aachen University and Roko Zaja, Institute of Biochemistry and Molecular Biology, RWTH Aachen University

Review Timeline:

Submission Date:	2022-03-18
Editorial Decision:	2022-04-22
Revision Received:	2022-09-16
Editorial Decision:	2022-10-13
Revision Received:	2022-10-18
Accepted:	2022-10-19

Scientific Editor: Novella Guidi

Transaction Report:

April 22, 2022

Re: Life Science Alliance manuscript #LSA-2022-01455

Dr. Karla Feijs
Institute of Biochemistry and Molecular Biology, RWTH Aachen University
Pauwelsstrasse 30
Aachen 52074
Germany

Dear Dr. Feijs,

Thank you for submitting your manuscript entitled "Protein and RNA ADP-ribosylation detection is influenced by sample preparation and reagents used" to Life Science Alliance. The manuscript was assessed by expert reviewers, whose comments are appended to this letter. We invite you to submit a revised manuscript addressing the Reviewer comments.

Thank you for this interesting contribution to Life Science Alliance. We are looking forward to receiving your revised manuscript.

Sincerely,

B. MANUSCRIPT ORGANIZATION AND FORMATTING:

Reviewer #1 (Comments to the Authors (Required)):

In the manuscript the authors tested and compared several reagents in the field of mono-ADP-ribosylation (MARylation) on various well defined substrates. In my eye this is a very carefully conducted and needed study that highlights the importance of careful sample preparation to avoid both false-negative and -positive MARylation signals. I agree with the authors that the study provides a comparison of the currently available reagents, which will allow researchers to make an informed decision as to which reagent to use for their specific purposes. Therefore, I recommend publication as it stands.

Reviewer #2 (Comments to the Authors (Required)):

This is a useful resource paper that will be very helpful to biochemists working with ADPRibosylation. It is a thorough study comparing several ADPR reagents that are used by researchers in the field. There are some interesting observations like how these reagents can pick up ADPRibylation signals on RNA. The data quality is good and with some modifications the manuscript can be published.

Some specific comments to improve the manuscript is as follows:

- The authors may consider briefly mentioning the source of the reagents in the introduction instead of mentioning them as references
- What is the mutant mentioned in Fig 1b?
- It will be useful to provide the table with description/source of the reagents in the figure where they are first used instead of the methods section
- It will be good to test the pH sensitivity of the ADPR signal in slot-blot of Fig 3f in addition to heating at 60deg and 90 deg.
- The authors should quantitate nuclear to cytosolic ADPR signal in your images in fig 6 to visualize better how the staining pattern changes based on fixation method and H₂O₂ treatment. The authors may check how saponin permeabilisation (0.1% saponin instead of triton-x) affects the staining pattern since it is less harsh than TritonX.
- Authors can add a summary table at the end to compare the 7 reagents in the different assays performed, if possible mentioning which is the best reagent for each assay

Reviewer #3 (Comments to the Authors (Required)):

In the manuscript titled " Protein and RNA ADP-ribosylation detection is influenced by sample preparation and reagents used" Weixler et al compare and contrast reagents for ADP-ribosylation analysis. Given the emerging role of ADP ribosylation in cell signaling, this paper provides a valuable resource that describes the enzymes that act as writers, eraser and readers in addition to the antibody reagents. The authors provide thorough description of purification schemes, lysis conditions and sample preparation for the transferases and hydrolases. I recommend this manuscript for publication with the following revisions:

1. Please provide Coomassie stained images for the proteins in panels 1b to 1e to show equal loading.
2. Data for MacroD2 is not shown in Fig 1 as mentioned in line 142.
3. Polymers of ADPr are seen with in the C3bot lane with Reagent 1 and par10cat with Reagent 4. This is in contrast to line 199 to 200. Please clarify if the observed smear corresponds with poly-ADPr or non- specific bands.
4. Line 205: Reagent 1 is not shown in Fig 2i
5. To address line 393, the authors could treat the western blot membranes with hydroxylamine to determine the specificity of reagents and IF signal.
6. Please briefly address the differences in slot blot vs western blot (electrophoresis) when analyzing signal. For example, the loss in signal that occurs when boiling samples may occur due to aggregation of protein that is stuck in the well and may not migrate into the resolving gel.
7. In response to line 263: In Fig 1b-d and 3c, the activity of the inactive mutant may be compared to demonstrate the specificity of the respective hydrolase or transferase.

Reviewer #1 (Comments to the Authors (Required)):

In the manuscript the authors tested and compared several reagents in the field of mono-ADP-ribosylation (MARylation) on various well defined substrates. In my eye this is a very carefully conducted and needed study that highlights the importance of careful sample preparation to avoid both false-negative and -positive MARylation signals. I agree with the authors that the study provides a comparison of the currently available reagents, which will allow researchers to make an informed decision as to which reagent to use for their specific purposes. Therefore, I recommend publication as it stands.

We appreciate that the reviewer took the time to evaluate our work and are glad that they find our work useful.

Reviewer #2 (Comments to the Authors (Required)):

This is a useful resource paper that will be very helpful to biochemists working with ADPribosylation. It is a thorough study comparing several ADPR reagents that are used by researchers in the field. There are some interesting observations like how these reagents can pick up ADPribylation signals on RNA. The data quality is good and with some modifications the manuscript can be published.

We would like to thank the reviewer for assessing our work and for the positive comments.

Some specific comments to improve the manuscript is as follows:

- The authors may consider briefly mentioning the source of the reagents in the introduction instead of mentioning them as references

We have added both the names of the labs' pioneering certain reagents, as well as the companies which now provide them commercially.

- What is the mutant mentioned in Fig 1b?

This is a MACROD1 glycine 270 to glutamate mutant, which blocks catalytic activity. We have adjusted the figure labeling to make this clear.

- It will be useful to provide the table with description/source of the reagents in the figure where they are first used instead of the methods section

We have integrated a small table in the text, which lists the source and type of reagents used.

- It will be good to test the pH sensitivity of the ADPR signal in slot-blot of Fig 3f in addition to heating at 60deg and 90 deg.

We have performed and added an additional experiment, where we tested stability in neutral, basic and acidic buffers (new supplementary figure 6). This experiment shows that a neutral pH is in general best, with different sensitivities of the different linkages especially to acidic pH.

- The authors should quantitate nuclear to cytosolic ADPR signal in your images in fig 6 to visualize better how the staining pattern changes based on fixation method and H₂O₂ treatment. The authors may check how saponin permeabilisation (0.1% saponin instead of triton-x) affects the staining pattern since it is less harsh than TritonX.

We have performed an additional experiment (new figure 6d) where we crosslinked diverse metabolites to BSA using PFA and tested the reagents on these substrates. NADH can be crosslinked to BSA and some reagents detect this metabolite. We think that the cytosolic staining visible after PFA fixation is caused by a metabolite. Upon H₂O₂ treatment, all signal is present in the nucleus and in unstimulated cells, the overall signal is very low. In MeOH-fixed cells, no cytosolic staining is present.

This corresponds with the western blots, where in untreated cells only very low amounts of ADP-ribosylation are visible. We have adjusted the discussion accordingly. While in general it is possible to also try saponin, we feel this is beyond the scope of our current work, especially considering that all antibodies that do not recognise NADH, detect no ADP-ribosylation under basal conditions.

- Authors can add a summary table at the end to compare the 7 reagents in the different assays performed, if possible mentioning which is the best reagent for each assay

We have added a table with a short summary of the reagents, which highlights their strengths and potential weaknesses.

Reviewer #3 (Comments to the Authors (Required)):

In the manuscript titled " Protein and RNA ADP-ribosylation detection is influenced by sample preparation and reagents used" Weixler et al compare and contrast reagents for ADP-ribosylation analysis. Given the emerging role of ADP ribosylation in cell signaling, this paper provides a valuable resource that describes the enzymes that act as writers, eraser and readers in addition to the antibody reagents. The authors provide thorough description of purification schemes, lysis conditions and sample preparation for the transferases and hydrolases. I recommend this manuscript for publication with the following revisions:

We would like to thank the reviewer for reading our paper and for indicating that it is a valuable resource for the community.

1. Please provide Coomassie stained images for the proteins in panels 1b to 1e to show equal loading.

The requested Coomassie stainings are available as supplementary figure 2.

2. Data for MacroD2 is not shown in Fig 1 as mentioned in line 142.

We indeed did not test MACROD2 here, and have removed this from the text.

3. Polymers of ADPr are seen with in the C3bot lane with Reagent I and par10cat with Reagent 4. This is in contrast to line 199 to 200. Please clarify if the observed smear corresponds with poly-ADPr or non- specific bands.

We have adjusted the indicated text. As the ADP-ribosylating toxins have very defined substrates, we assume the larger bands present with Reagent I are unspecific. The PARP10cat we used, is highly active and may possibly modify some contaminant proteins present at very low levels in the reactions. We have adjusted the text to describe this ambiguity better.

4. Line 205: Reagent 1 is not shown in Fig 2i

We are grateful to the reviewer for detecting this mistake and have corrected it.

5. To address line 393, the authors could treat the western blot membranes with hydroxylamine to determine the specificity of reagents and IF signal.

We have performed a different, additional experiment (new figure 6d), wherein we crosslinked NAD⁺, NADH or ADPr to BSA and have slot-blotted this. It appears that PFA efficiently crosslinks NADH, which is then detected by several antibodies. The mitochondrial staining observed in IF is thus most likely not a modified protein, but NADH.

6. Please briefly address the differences in slot blot vs western blot (electrophoresis) when analyzing

signal. For example, the loss in signal that occurs when boiling samples may occur due to aggregation of protein that is stuck in the well and may not migrate into the resolving gel.

We have added a paragraph to explain better the difference between western and slot blotting.

7. In response to line 263: In Fig 1b-d and 3c, the activity of the inactive mutant may be compared to demonstrate the specificity of the respective hydrolase or transferase.

We agree that inactive mutants would be useful to further test this hypothesis in cells. For the in vitro assays, the transferases' substrates are relatively well defined and are reversed by the expected hydrolases, indicating that the ADP-ribosylation is present on the anticipated amino acids. Testing whether expression of inactive transferases in cells might lead to activation of other enzymes however is beyond the scope of the current resource article and might make an interesting topic for future work. We have emphasized this possibility more clearly in the revised text.

October 13, 2022

RE: Life Science Alliance Manuscript #LSA-2022-01455R

Dr. Karla Feijs
Institute of Biochemistry and Molecular Biology, RWTH Aachen University
Pauwelsstrasse 30
Aachen 52074
Germany

Dear Dr. Feijs,

Thank you for submitting your revised manuscript entitled "Protein and RNA ADP-ribosylation detection is influenced by sample preparation and reagents used". We would be happy to publish your paper in Life Science Alliance pending final revisions necessary to meet our formatting guidelines.

- please upload both your main and your supplementary figures as single files
- please add ORCID ID for secondary corresponding author-they should have received instructions on how to do so
- please add a conflict of interest statement to your main manuscript text
- please consult our manuscript preparation guidelines <https://www.life-science-alliance.org/manuscript-prep> and make sure your manuscript sections are in the correct order
- please add a separate section for your main and supplementary figure legends and add a legend for figure S2.
- please use the [10 author names, et al.] format in your references (i.e. limit the author names to the first 10)

Figure Check:

- please add molecular weights next to all blots
- please upload figure S9 as Source Data
- Figure 1F: please provide source data for this figure
- Figure duplications are not allowed at our journal. Here below are the duplications found: Fig 1B=Fig S2A, Fig 1C=Fig S2B, Fig 1D=Fig S2C, Fig 1E=Fig S2E. Please remove the duplicate figures from supplementary. You are always welcome to provide source data images for Figure 1 that are not identical to the panels in Figure 1
- Figure 2I: please provide source data for this figure
- There is a partial duplicate in Figure 3d, 4th and 5th columns, with Figure 4B: please provide source data for both figure 3D and 4B
- Figure 6 A,B,C: please add scale bars
- it seems that Figure S4 contains source data images. If this is the case, please provide figure S4 as source data

A. FINAL FILES:

- An editable version of the final text (.DOC or .DOCX) is needed for copyediting (no PDFs).
- High-resolution figure, supplementary figure and video files uploaded as individual files: See our detailed guidelines for

preparing your production-ready images, <https://www.life-science-alliance.org/authors>

B. MANUSCRIPT ORGANIZATION AND FORMATTING:

Sincerely,

Reviewer #2 (Comments to the Authors (Required)):

The authors have addressed all the reviewer concerns and have added experiments wherever needed. According to my opinion this is a nice useful study and can be accepted for publication in its present form.

Reviewer #3 (Comments to the Authors (Required)):

The authors have incorporated changes as suggested by the reviewers. The manuscript is suitable for publication.

October 19, 2022

RE: Life Science Alliance Manuscript #LSA-2022-01455RR

Dr. Karla Feijs
Institute of Biochemistry and Molecular Biology, RWTH Aachen University
Pauwelsstrasse 30
Aachen 52074
Germany

Dear Dr. Feijs,

Thank you for submitting your Resource entitled "Protein and RNA ADP-ribosylation detection is influenced by sample preparation and reagents used". It is a pleasure to let you know that your manuscript is now accepted for publication in Life Science Alliance. Congratulations on this interesting work.

DISTRIBUTION OF MATERIALS:

Again, congratulations on a very nice paper. I hope you found the review process to be constructive and are pleased with how the manuscript was handled editorially. We look forward to future exciting submissions from your lab.

Sincerely,
